# Dietary Risk Factors and Eating Behaviors in Peripheral Arterial Disease (PAD)

**DOI:** 10.3390/ijms231810814

**Published:** 2022-09-16

**Authors:** Andrea Leonardo Cecchini, Federico Biscetti, Maria Margherita Rando, Elisabetta Nardella, Giovanni Pecorini, Luis H. Eraso, Paul J. Dimuzio, Antonio Gasbarrini, Massimo Massetti, Andrea Flex

**Affiliations:** 1Internal Medicine, Università Cattolica del Sacro Cuore, Largo Francesco Vito 1, 00168 Rome, Italy; 2Cardiovascular Internal Medicine, Fondazione Policlinico Universitario A. Gemelli IRCCS, Largo Agostino Gemelli 8, 00168 Rome, Italy; 3Division of Vascular and Endovascular Surgery, Thomas Jefferson University, Philadelphia, PA 19107, USA; 4Department of Medical and Surgical Sciences, Fondazione Policlinico Universitario A. Gemelli IRCCS, Largo Agostino Gemelli 8, 00168 Rome, Italy; 5Department of Cardiovascular Sciences, Fondazione Policlinico Universitario A. Gemelli IRCCS, Largo Agostino Gemelli 8, 00168 Rome, Italy

**Keywords:** peripheral artery disease of lower limbs, lower extremity arterial disease, food, eating behaviors, eating disorders, nutrition, predictors, dietary risk factors, diet

## Abstract

Dietary risk factors play a fundamental role in the prevention and progression of atherosclerosis and PAD (Peripheral Arterial Disease). The impact of nutrition, however, defined as the process of taking in food and using it for growth, metabolism and repair, remains undefined with regard to PAD. This article describes the interplay between nutrition and the development/progression of PAD. We reviewed 688 articles, including key articles, narrative and systematic reviews, meta-analyses and clinical studies. We analyzed the interaction between nutrition and PAD predictors, and subsequently created four descriptive tables to summarize the relationship between PAD, dietary risk factors and outcomes. We comprehensively reviewed the role of well-studied diets (Mediterranean, vegetarian/vegan, low-carbohydrate ketogenic and intermittent fasting diet) and prevalent eating behaviors (emotional and binge eating, night eating and sleeping disorders, anorexia, bulimia, skipping meals, home cooking and fast/ultra-processed food consumption) on the traditional risk factors of PAD. Moreover, we analyzed the interplay between PAD and nutritional status, nutrients, dietary patterns and eating habits. Dietary patterns and eating disorders affect the development and progression of PAD, as well as its disabling complications including major adverse cardiovascular events (MACE) and major adverse limb events (MALE). Nutrition and dietary risk factor modification are important targets to reduce the risk of PAD as well as the subsequent development of MACE and MALE.

## 1. Introduction

Our planet is currently inhabited by approximately 7850 billion people. With a growth rate of 1.05% per year, the average population increase is estimated at 81 million people yearly (World Population Clock—2021). Meanwhile, cardiovascular disease (CVD) is the leading cause of death globally resulting in approximately 17.9 million deaths annually (World Health Organization—WHO). Despite the enormity of cardiovascular disease and the efforts to promote the importance of its prevention, nutrition remains a dramatically underestimated aspect in addressing this “atherosclerosis pandemic” [1].

Healthy diets can play an essential role in preventing and/or delaying major atherosclerotic complications. For example, diets high in sodium yet low in whole grains, nuts, seeds, vegetables, omega-3 fatty acids and fruits have recently been identified as the main dietary risk factors responsible for 10 million deaths from CVD and 207 million cardiovascular diseases worldwide [2,3]. Furthermore, eating behaviors have recently been shown to be relevant predictors of poor CVD outcomes [4].

Peripheral Artery Disease (PAD) of the lower limbs is a disabling complication of atherosclerosis and shares a common etiology with coronary artery and cerebrovascular disease [5]. Patients with PAD exhibit more aggressive multivessel atherosclerotic burden [6] and often have multiple risk factors and comorbidities that promote the progression of this disease [7].

Given that nutrition and dietary patterns are poorly studied aspects of the management of these patients with PAD, [1] we studied how adherence to anti-atherogenic diets and common eating habits/disorders may affect several important risk factors for PAD. We present an extensive review of the literature on the relationship between PAD and nutrition, and subsequently suggest nutritional recommendations specific to individual needs and characteristics [8].

### 1.1. Materials, Methods and Study Design

This narrative review analyzes the risk factors for PAD as they relate to nutrition, food, diet and eating behaviors. We entered selected keywords into both PubMed and Google Scholar to identify the appropriate literature; a list of the selected keywords is in the “keywords section”. Ultimately, we selected 688 articles published between 1980 and 2022, including key articles, narrative and systematic reviews, meta-analyses and clinical studies, for analysis. The review was divided into three main sections including: (1) an introduction, (2) a description of the impact of nutrition on predictors of PAD and (3) the interplay between nutrition and PAD (Figure 1). The main results are summarized in four descriptive tables (Table 1, Table 2, Table 3 and Table 4).

### 1.2. Non-Dietary Risk Factors and Predictors of Peripheral Artery Disease

Atherosclerosis is a chronic, progressive disease that affects the main arterial beds including the coronary, carotid and peripheral arterial trees [461]. The involvement of the lower limb vessels defines PAD and patients with this condition are often complex and fragile given their significant cardiovascular risk [462]. PAD shares the pathological substrate with the other manifestations of atherosclerosis and is responsible for disabling complications that affect their quality of life and survival [463,464].

An important aspect of PAD is the study of the risk factors underlying the disease and influencing patient outcomes [465]. Among the non-dietary risk factors of PAD, obesity [466,467,468,469,470,471], smoking [472,473,474,475,476], diabetes mellitus [10,477,478,479,480], chronic kidney disease [481,482,483,484,485,486,487,488], hypertension a [478,480,489,490,491,492], dyslipidemia [10,478,480,493,494,495,496,497] and systemic inflammation [498,499,500] play a significant role in the disease. Each of these are prevalent in patients with PAD and play a fundamental role in the disease progression. The treatment of concomitant comorbidities and the management of the risk factors affecting patients have proven to be an effective prevention strategy and therapeutic approach to improve individual morbidity and mortality [501].

Smoking cessation and the control of serum glucose levels are the primary goals for these patients, who are often long-time smokers and/or diabetics, to reduce oxidative vascular damage due to tobacco use or glucose toxicity [502]. Arterial calcification can be ameliorated by slowing the deterioration of renal function, which is responsible for the disturbance of calcium–phosphorus metabolism [503,504]. Adequate blood pressure management and the achievement of recommended cholesterol levels lead to minimizing the atherosclerotic burden [482,505,506,507,508].

Finally, “residual risk” involving systemic inflammation should be considered with regard to cardiovascular risk [509]. Many therapeutic strategies have been studied to directly treat inflammation; however, it appears that prevention against oxidative stress and the control of immune dysregulation are most effective in counteracting any subclinical inflammatory process [510].

## 2. Interplay between Risk Factors and Predictors of PAD with Dietary Patterns/Eating Behaviors

Recently, an epidemic of chronic diseases related to suboptimal eating behaviors and inadequate nutrition dramatically affected the mortality and morbidity of the global population [2]. One in five patients die due to the consequences of inappropriate diets on health, regardless of traditional cardiovascular risk factors [1,2]. Cardiovascular disease is the leading cause of death and morbidity attributable to poor nutrition, accounting for approximately 10 million deaths and 207 million disability-adjusted life years (DALYs) each year [2]. Among those comorbidities related to dietary habits, we recognize several of the main risk factors for PAD, suggesting an interaction between nutrition, predictors of atherosclerosis and the development of PAD [465]. Furthermore, there is a high prevalence of the dietary risk factors in those suffering from chronic disease and exposed to multiple atherosclerotic risk factors. Among the most recognized dietary risk factors for overall mortality and morbidity—and those highly associated with atherosclerotic complications—are high sodium consumption, low intake of whole grains and fruit, as well as high intake of sugar (especially sugary drinks), red and processed meats and trans fats. Other easily recognizable nutritional aspects that parallel with cardiovascular risk include processes related to food manufacturing (processing, production, distribution, cooking), poor access to healthy food, insufficient supply of seeds, vegetables, omega-3 fatty acids and the adoption of unhealthy eating behaviors [511,512,513,514,515,516,517,518]. Healthcare systems should recognize the importance of including targeted diets and proper eating habits as a part of evidence-based therapy to improve the management of cardiovascular risk and patient outcomes [519,520,521,522,523,524,525,526].

### 2.1. Diets and Risk Factors/Predictors of Peripheral Artery Disease of Lower Limbs

A dietary scheme is a balanced composition of macro- and micronutrients that provides adequate caloric intake and benefits metabolism. Each diet has characteristics that can be tailored to the patient’s personal needs and comorbidities. We evaluated the relationship between various common and well-studied diets (Mediterranean diet, vegetarian and vegan diet, low-carbohydrates ketogenic diet, intermittent fasting) and the main risk factors of PAD (diabetes mellitus, obesity, hypertension, dyslipidemia, chronic kidney disease, inflammatory status). Current knowledge on each dietary model in relation to the selected risk factors and predictors of PAD are described, listing several pros and cons of each on patient comorbidity [527,528]. The main results are summarized in Table 1.

#### 2.1.1. The Mediterranean Diet

The Mediterranean diet (MD) has received particular scientific interest in recent years as it induces a significant reduction in CV risk via a balanced composition of macronutrients (carbohydrates, proteins and fats) [529,530]. The most important characteristics of the MD are: (1) the moderate consumption of lean meat and fish with a minimum intake of red or processed meat, (2) the voidance of sugary drinks, (3) a moderate intake of salt and dairy products (especially cheese and yogurt) and (4) an abundance of vegetables, seeds, legumes (e.g., lentils and beans), fruit, cereals and whole grains (e.g., unprocessed maize, millet, oats, wheat and brown rice) [531]. Unlike a Western diet, the MD reduces the consumption of saturated fats (almost avoiding products such as butter) by including foods rich in unsaturated fats (mono- and polyunsaturated) such as olive oil, nuts and seeds, used as main courses or cooking ingredients [532]. Extra virgin olive oil and red wine, rich in polyphenols, tocopherols and phytosterols, provide anti-inflammatory characteristics and valuable cardiovascular protection properties, such that they are considered pillars of the MD [533,534,535]. The diet’s benefits are recommended as part of the preventive and therapeutic strategy in patients at a higher cardiovascular risk given their documented reduction in overall mortality [128].

The MD also includes appropriate eating and non-eating behaviors [536,537] that may ameliorate the ongoing obesity and diabetes pandemic [148]. The Mediterranean lifestyle and eating habits are effective solutions to the harmful consequences of a “Westernization” of life, including incorrect eating behaviors and physical inactivity, that are responsible for the higher prevalence of chronic diseases, especially diabetes and obesity [149,150,151,152,153,154]. The MD counteracts weight gain [155] by changing intestinal microbiota (e.g., the Firmicutes/Bacteroidetes ratio), increasing energy expenditure via the thermogenesis of brown adipose tissue and inducing lipolysis [156]. In particular, the healthy composition of nutrients in the MD is key to decreasing the incidence of diabetes and obesity. In fact, the MD includes a large variety of plant-based foods, polyunsaturated fats, fruits, whole grain products, fish and fiber together with a reduced intake of processed and red meats, refined sugars and saturated fats [162,530]. Weight loss is also facilitated through the regulation of satiety promoted by the consumption of products rich in short-chain fatty acids that induce the production of incretin and the associated control of blood sugar and insulin sensitivity [157]. Furthermore, the improvement of insulin resistance can be explained by a lower intake of carbohydrates with a high glycemic index and an increase in the intake of monounsaturated acids, essential and branched-chain amino acids that favor glucose control [158], preventing the development and progression of diabetes [159,160,161]. Additionally, those who adopt the MD have greater adherence to health-promoting behaviors (such as getting enough sleep, better education, higher incomes, more physical activity and less smoking) that affect food quality, susceptibility to weight gain and the prevention of several comorbidities [530,538,539,540,541].

The MD has documented beneficial effects on hypertension. The reduction in saturated fats replaced by olive oil or mixed nuts in the MD results in a significant reduction in blood pressure [162,163,164]. A stricter adherence to the MD has been shown to have additional benefit in the prevention and treatment of hypertension along with the traditional pharmacological treatment [164]. The MD plays a key role in reducing cardiovascular risk by preventing the development and controlling the main chronic diseases that increase the incidence of adverse vascular events [165,166].

In addition to a reduction in blood pressure and the overall risk of mortality/morbidity, the MD may reduce arterial stiffness [167] and endothelial dysfunction [168], known conditions related to atherosclerosis. Nutrients included in the diet may alter various molecular processes that slow down the vessel degeneration observed in atherosclerosis. These include: (1) increased arterial dilation mediated by a higher production of nitric oxide, (2) a reduced expression of Caveolin-2 [169,170] and endothelin-1 [171], (3) the down regulation of the JUN gene pathway [172], (4) the inhibition of the NF-kB/AP-1 signal and ADMA responsible for a reduction in the bioavailability of nitric oxide and (5) the modulation of the adrenergic nervous system [173] mainly mediated by oleic acid, monounsaturated fatty acids and polyphenols.

The MD is included in the therapeutic recommendations of the non-pharmacological management of atherosclerotic disease since the heterogeneous composition of nutrients and the quality of foods found in the MD [158] play an important role in the regulation of lipid metabolism with a documented reduction in the overall CV risk [174]. It is an effective dietary strategy in the prevention of dyslipidemia and could prove to be a successful approach to achieve the recommended therapeutic goals for cholesterol. The diet has a direct effect on the serum lipid profile with a reduction in levels of cholesterol, triglycerides and atherogenic apolipoproteins along with an increase in serum HDL-c [175]. The abundance of plant-derived foods, olive oil and nuts, along with a low intake of processed foods, dairy products and red meats are examples of MD recommendations that have lipid-lowering effects with a consequent reduction in atherosclerosis [176,177], especially in patients with comorbidities [178]. Additionally, the low consumption of foods rich in saturated fats espoused by the MD results in a decreased endogenous production of cholesterol [179]. The MD also offers a wide choice of foods that directly lower cholesterol levels such as the polyunsaturated fats of vegetable origin, olive oil, seeds, nuts, vegetables and fruit. The water-soluble fibers in fruit and legumes have a lipid-lowering effect by reducing the intestinal absorption of cholesterol and bile acids, promoting the hepatic uptake of LDL-c.

Complex carbohydrates and fibers (such as cereals and whole meal pasta) are low glycemic index foods that contribute to intestinal fermentation, modulate insulin production and lead to a greater synthesis of short-chain fatty acids with a consequent reduction in serum cholesterol levels [180,181]. Phytosterols are plant-based fats, similar to cholesterol. These molecules compete with the intestinal absorption of cholesterol which favors its elimination, directly improving the cardiovascular risk by favorably influencing the lipid profile [182,183]. Currently, the heterogeneous inter-individual response to dietary patterns is a new topic of interest and the promising results on dyslipidemia observed in people with a strong family history of hypercholesterolemia suggests a possible epigenetic regulation of lipid metabolism by the MD [184]. Long-term adherence and the early adoption of the diet, especially from preschool life [185], exert the most significant effects on health, cardiovascular protection and the lipid profile [186,187,188], confirming the importance of precocious nutritional strategies for primary prevention. However, a well-structured nutritional model that includes all pillars of the MD in primary and secondary cardiovascular prevention should be tested in clinical practice and new studies are needed to further expand the mechanism underlying the benefits of adopting a lifestyle and a Mediterranean dietary model in atherosclerotic diseases [542,543,544].

Chronic kidney disease is characterized by an irreversible and progressive decline in kidney function, which determines a profound modification of cardiometabolic homeostasis. Renal failure is responsible for the accumulation of various uremic toxins derived from the intestinal metabolism of foods rich in GDUTs (uremic toxins of intestinal origin). The MD was expected to reduce the serum levels of trimethylamine N-oxide (TMAO), p-cresyl sulfate, hippuric acid, noxyl sulfate, p-cresyl glucuronide, phenyl acetyl glutamine and phenyl sulfate by a moderate-to-low intake of typical products that include the precursors of GDUTs such as milk, eggs, meat and dairy foods. However, even strict adherence to the MD does not seem to prevent these pro-inflammatory and atherogenic toxins from accumulating in the serum of patients with severe renal insufficiency [189]. Overall, there is still no clear evidence of the protective role of the MD on the progression of renal disease [190].

The MD, however, includes many components that may delay the progression of renal failure defined as eGFR <60 mL/min [191,192,193,194,195]. Although the effect of diet on renal function is unclear, a lower incidence of mortality was observed among chronic kidney disease (CKD) patients who adopted dietary recommendations similar to that of the MD [196]. Patients affected by chronic renal failure who adhere to the MD demonstrate a survival advantage over the controls due to a better nutritional profile, as demonstrated by higher levels of hemoglobin and albumin and lower levels of pro-inflammatory molecules [198]. Further, it has been observed that kidney transplant recipients have a lower incidence of transplant failure, decline in kidney function and urinary protein excretion when on the MD [199]. The consumption of fruits and vegetables, ensuring good hydration and minimizing the intake of animal proteins, salt and dairy products prevents the formation of kidney stones [200]. Finally, the safety of the MD in CKD is unresolved due to the high content of foods rich in minerals such as legumes, nuts, fruits and vegetables (sources of potassium and phosphorus). The cooking method seems to reduce mineral accumulation in the blood in the case of kidney disease; however, several concerns limit the safe application of the MD in CKD patients [201,202]. While the MD may benefit patients with CKD, there remains safety and efficacy in this patient population [191,203,204,205].

Atherosclerosis is a systemic inflammatory process marked by the increased production of cytokine and immune dysregulation, and renders the patient susceptible to cardiovascular events [545]. The MD is a “hormetic therapy” because it includes several nutrients which restore the physiological homeostasis of the compromised immune system in patients with advanced atherosclerosis [206]. The diet is rich in anti-inflammatory products such as polyphenols (mainly contained in olive oil) and foods of plant origin (such as legumes, fruit, vegetables, dried fruit); conversely, it espouses a low consumption of saturated fatty acids and meat (with preference of lean meat) which promote inflammation [207,208,209]. As such, the nutritional recommendations in the MD are effective in addressing this new risk factor, especially if treatment adherence is maintained from an early age [210]. Exploring this interaction further, the MD modulates the inflammatory state by acting on several levels. At the molecular level, it reduces adipose tissue deposits and consequently lowers the release of pro-inflammatory cytokines [211,212]. Moreover, lower levels of IL-6, TNFα, CRP, adhesion molecules (such as ICAM-1) and other oxidative stress markers [166,213,214,215,216,217,546,547], as well as high levels of adiponectin and other anti-inflammatory cytokines (such as IL-10), have been documented [197]. The MD can also regulate fundamental biochemical and molecular pathways involved in systemic inflammation through pre-transcriptional genomic and epigenomic modifications (e.g., histone deacetylation, DNA methylation, the regulation of miRNAs) [158,218]. Extra virgin olive oil is a key element in the MD and plays an important role in dysbiosis by restoring the composition of the intestinal microbiota and directly influencing host immunity [214,217,219,220,221,548]. Furthermore, the abundance of polyphenols in the MD can exert health benefits directly in the atherosclerotic plaque by modulating the local immune system, restoring the integrity of the vessels and partially slowing down endothelial dysfunction, which are preliminary steps in the atherosclerotic process [222].

Thus, evidence suggests that the MD minimizes the harmful consequences of maladaptive and dysregulated inflammation in atherosclerosis by restoring the balance between proinflammatory mediators [186] and anti-inflammatory responses [223,224,225].

The mortality and morbidity benefits documented after the introduction of the MD in patients at high cardiovascular risk are in part be due to better control of the inflammatory profile responsible for the progression of several chronic diseases [226], yet inflammation remains an untreated residual risk factor [197]. Given the great success in managing the inflammatory profile of patients, the MD is recommended by both the ACC/AHA and ESC/EAS guidelines as an integral part of non-drug therapy to counteract the risk and incidence of major adverse vascular events.

#### 2.1.2. The Vegetarian, Vegan and Plant-Based Diet

The minimal consumption of animal products has shown sound health benefits. Adopting a vegetarian diet, or completely avoiding proteins of animal origin as in a whole plant-based diet, have documented protection against the development of major cardiovascular risk factors with a modest increase in longevity [99,107,110].

The adoption of vegetarian and vegan diets demonstrated a much lower prevalence of obesity compared to the consumption of a diet based on animal proteins, as demonstrated in the epidemiological analysis [99,100]. Plant-based diets are rich in water, complex carbohydrates and fibers that promote an earlier and long-term satiety with an absolute increase in resting energy expenditure. In the diet, high calorie foods are replaced by those that benefit the gut microbiota composition, which in turn leads to lower levels of trimethylamine-N-oxide (TMAO), increases insulin sensitivity, activates peroxisome proliferator-activated receptors (PPARs) [549] and ameliorates mitochondrial pathways [101]. Vegetarians and vegans tend to be have a lower BMI and have additional protection against the development of obesity [100,102], confirming that the reduction in animal protein consumption is a preventive strategy against obesity [102,103]. The low consumption of animal proteins appears to improve adipocyte function. An interesting and relatively new aspect is the effect of a vegan diet on adipokines, which have been observed as a promising target for counteracting the metabolic syndrome [550]. In particular, replacing animal proteins with mixed gluten–soy proteins and soy proteins may increase serum adiponectin levels, influence AdipoR1 mRNA expression in skeletal muscles and promote adiponectin production in adipose tissues [551]. Furthermore, lacto-ovo vegetarian and vegan diets showed a significant reduction in serum levels of leptin, perhaps secondary to the high availability of PUFA and the low consumption of animal proteins, which directly affect the production of this hormone in adipose tissue. The effect of vegetarian and vegan diets on adipokines is not entirely clear but appears to contribute to the reduction in adiposity and excessive weight gain [552].

A whole plant-based diet with vitamin B12 supplementation has an impact on weight loss and body mass composition without documented serious harm [104,105]. A properly compiled plant-based diet may be a viable option to change nutritional habits to ameliorate the global obesity pandemic [106,107,108,109,110].

Plant-based diets decrease the risk of developing diabetes mellitus [99,111] by reducing insulin resistance and the impairment of serum glucose metabolism [112]. In addition to the preventive role, a veg nutrition has shown convincing results in the treatment of diabetes, with including a reduction in the use of hypoglycemic drugs [113,114,115]. Furthermore, the protective and therapeutic effect of a vegetarian/vegan diet appears independent of physical exercise [116]. Reducing the consumption of foods of animal origin decreases visceral fat and the production of adipokines, which contribute to oxidative stress. The transition to green nutrition improves the functionality of the pancreatic beta cells, the production of gastrointestinal incretins and the excretion and sensitivity to insulin, all of which results in greater control of diabetes [117,118,119,120]. Thus, a vegetarian or totally plant-based diet is a compelling strategy to prevent poor glucose control, achieving consistent results in the treatment of diabetes [121,122,123].

Patients with hypertension who adhere to a vegan/vegetarian diet achieve better blood pressure control [124]. This clinical benefit was independent of the extent of weight loss, reduction in potassium or salt intake and physical activity [125]. The protective role of a plant-based diet against the development/progression of hypertension was made in comparison to the blood pressure values of those on omnivorous diets. In fact, an adequate blood pressure profile and effective management of hypertension has an inverse relationship with the intake of animal origin products, while being directly proportional to compliance with adherence to an exclusively vegan diet [126]. Plant-based diets have a safe profile and can be recommended in both the prevention and non-drug therapy of hypertension, particularly for those patients who demonstrate good compliance and need additional support to achieve blood pressure goals [126].

Vegan and vegetarian diets benefit the lipid profile and can be considered an anti-atherogenic therapy given their impact on dyslipidemia and overall cardiovascular risk. On these diets, patients realize a significant reduction in serum cholesterol and apolipoproteins that are associated with the progression of atherosclerotic plaque [127,128]. Additionally observed is an improvement in LDL-C serum levels due to the lower consumption of saturated fats [128]. Foods of animal origin (such as meat, dairy products and eggs) increase LDL-C concentrations and subsequently cardiovascular risk and major adverse events [129]. Surprisingly, a reduction in animal proteins and fats from one’s diet has an impact on lipid metabolism comparable to the effect of statins [128,130], with further benefit noted with regular physical activity and smoking cessation [131]. Adherence to a veg diet is responsible for a significant loss of visceral fat (including hepatocellular and intramyocellular fat), increased calorie expenditure and improved serum lipid profile with a significant reduction in triglycerides and LDL-c along with an increase in HDL-C levels [132]. The remarkable improvements in both fasting and post-prandial blood lipids should encourage the introduction of balanced vegetarian and vegan diets in patients with a high cardiovascular risk and dyslipidemia [110].

The adoption of a vegetarian/vegan diet may help prevent kidney disease and protect patients from a deteriorating glomerular filtration rate [133]. The elimination of animal proteins by adopting an entire plant-based diet reverses the deleterious effects of the nitrogen content of meat on renal function. A significant reduction in products of animal origin regulates the systemic inflammatory process by modulating gut microbiota composition, decreasing renal hyperfiltration secondary to excessive protein intake and improving the ability to buffer an acidic environment that may facilitate inflammation and accelerate kidney disease [133]. The renal protective effect is more effective as the patient minimizes the intake of products of animal origin, which increase the nitrogen end-products that worsen uremia. A diet consisting of at least half meals based on plant-based foods significantly slows the progression of CKD by protecting nephrons from hyperfiltration damage due to intra-glomerular pressure [134]. An increase in fruits, vegetables and legumes could lead to an excessive potassium intake, exposing patients with chronic renal failure to develop hyperkalemia; however, the greater intake of fiber compensates by reducing constipation and the risk of a dangerous increase in serum potassium [134]. Predominantly plant-based diets improve uremic symptoms and ameliorate complications including metabolic acidosis, hypertension, systemic inflammation, proteinuria, mineral disturbances and the need for dialysis [134,135,136,137,553]. The diet may stabilize kidney disease and improve outcomes in CKD patients due to pleiotropic effects; however, simultaneous medical and nutritional evaluations are recommended to avoid malnutrition and hyperkaliemia in CKD patients [133,136].

The high intake of fruits, vegetables, legumes and fiber provides antioxidants, which counteract systemic oxidative stress, reduce the chronic low-grade inflammation observed in atherosclerosis and prevent the development of its complications [138,139,140,141]. A substantial decrease in inflammatory cytokine production and immune cell activity is documented in plant-based diets compared to omnivorous diets [139]. The progressive reduction in the intake of animal proteins and fats is associated with lower inflammatory biomarkers such as C-reactive protein, [142,143], lipoproteins responsible for the increase in serum levels of low-density LDL-c particles, leukocytes, interleukin-6 (IL-6) and TGF- β [141,144]. The interaction between vegetarian/vegan nutrition and the immune system [139] starts in the gut. In fact, the transition from a diet rich in saturated fat, cholesterol and iron content to a diet that provides a greater intake of fiber, antioxidants, polyunsaturated fats and micronutrients results in a net reduction in the inflammatory profile linked to obesity and restores the intestinal homeostasis of the microbiota [141,145]. The adoption of a plant-based food model significantly influences the intestinal environment and the composition of the microbiome, favoring the selection of anti-inflammatory bacteria and reducing the production of pro-inflammatory cytokines triggered by gut dysbiosis (including the altered Firmicutes/Bacteroidetes ratio) [146,147]. The anti-inflammatory properties of lacto-ovo vegetarian or whole plant-based diets should be considered a therapeutic strategy to reduce the inflammatory triggers underlying the progression of atherosclerosis.

#### 2.1.3. The Low-Carbohydrate Ketogenic Diet

A low-carbohydrate ketogenic diet drastically reduces carbohydrate intake while increasing protein and fat consumption. The purpose is to steer metabolism towards a greater consumption of fat as a source of energy. The accumulation of acetyl-CoA due to increased fat oxidation, together with low oxaloacetate production, leads to the formation of ketone bodies in the mitochondrial matrix of liver cells (acetoacetate (AcAc), β-hydroxybutyric acid (BHB) and acetone) [554].

The diet results in a rapid reduction in body weight due to a marked caloric waste and higher energy expenditure [11]. Ketogenesis leads to a significant consumption of calories that leads to gluconeogenesis [12], induces thermal expenditure secondary to protein metabolism [12,13] and promotes fat oxidation through the lipolysis process [14,15,16]. Furthermore, it suppresses hunger due to the sense of satiety provided by proteins and ketone bodies [17,18,19,20].

Loss of weight and fat mass directly benefits diabetic management [42], resulting in improved glycemic control with an expected reduction in glycated hemoglobin [41]. This progressive glucometabolic compensation can also result in a reduction in insulin therapy requirements, and in some cases, leads to the suspension of pharmacological treatment [21,22] due to a reduction in insulin resistance regardless of the extent of weight loss [23,24,25]. The restriction of carbohydrates and the consequent benefits to some fundamental metabolic pathways involved in the development of diabetes (such as insulin-like growth factor-1, phosphoinositide 3-kinase, protein kinase B and mammalian rapamycin) make KD an effective strategy for glycemic control and diabetes prevention [26,27]. The long-term safety of a low-carbohydrate/ketogenic diet is an unresolved issue that requires further investigation [555,556] for safe application in obese and diabetic patients [557].

The impact of a ketogenic diet on the lipid profile is a topic of great interest and controversy, as the weight loss achieved by a drastic reduction in carbohydrates is associated with a compensatory increase in fat intake, including saturated fat [558]. Furthermore, rapid weight loss promotes an increase (or no reduction) in LDL-c levels [559]. An increase in the size of the LDL-c molecule has been observed, but this does not seem to influence the risk of atherogenesis due to a lower trend of ectopic deposition in arterial walls [560,561].

The ketogenic diet has a direct effect on the lipid profile by increasing HDL-c levels and reducing serum triglycerides [25,35,36,37,38,39,40,41]. Therefore, through the regulation of the lipid profile and a reduction in the insulin-related activation of HMGCoA reductase (with the endogenous synthesis of cholesterol), the ketogenic diet seems to have a protective effect in atherogenic dyslipidemia with a documented structural and functional change in the LDL-c molecule [11,42].

In patients with hypertension, the ketogenic diet has demonstrated ambiguous results and requires further studies to understand its safety and efficacy. There is no evidence for blood pressure control [28] or hypertension’s harmful effects, including cardiac remodeling, endothelial dysfunction and the deterioration of arterial relaxation due to reduced nitric oxide synthase (NOS) production [29,30]. Patients with renal insufficiency appear particularly susceptible to increased blood pressure during the ketogenic diet, owing to acidosis caused by the metabolism of the amino acids involved in gluconeogenesis and increased urea production [11,31]. However, the protective role of the ketogenic diet in hypertensive patients is still possible [32] as groups of patients following KD have shown improved blood pressure profiles with decreased systolic and diastolic blood pressure values [33,34,35].

The use of the ketogenic diet in patients with chronic kidney disease is a recurring topic in the scientific community [562]. A low-carbohydrate diet often requires an increase in protein and fat to ensure a sufficient intake of calories and healthy nutrition. However, the increased ingestion of proteins, especially of animal origin, causes higher renal glomerular pressure, renal hyperfiltration, nephron damage and proteinuria, leading to CKD progression [563]. Furthermore, patients with chronic renal disease are more prone to altered mineral metabolism, and a ketogenic diet might expose patients to a greater risk of reduced bone mass, worsening the calcium–phosphorus homeostasis [43,564]. Although no direct renal benefits of a low carbohydrate ketogenic diet have been observed in patients with CKD, in cases of polycystic kidney disease, the metabolic shift from aerobic glycolysis to mitochondrial oxidative phosphorylation appears to reduce cyst growth [45] and promote renal cyst regression [46]. Notably, a ketogenic diet does not necessarily increase protein intake. In fact, an isocaloric ketogenic diet includes the greater ingestion of calories from other macronutrients such as fats [11,565,566]. Therefore, when KD is chosen as a therapeutic option for weight or blood glucose management in patients with CKD, supervision by health care professionals is appropriate [44].

The inflammation underlying multiple pathologies (including atherosclerosis) is a process induced by the chronic dysfunction of innate and adaptive immunity. Ketone bodies (especially β-hydroxybutyrate) can modulate the inflammasome [47] by reducing the macrophage production of interleukins such as IL-1β, IL-18 and TNF-α [48]. Ketone bodies also can decrease the activation of caspase-1 due to the regulation of the NLR family pyrin domain-containing 3, a sensor that detects damage-associated molecular products and is also activated by an excess of serum glucose and by atherosclerosis itself [567,568,569]. Furthermore, the effect of ketone bodies and caloric restrictions on the gut microbiota improves mitochondrial respiration, decreases oxidative stress and the production of reactive oxygen species (ROS), activates endogenous antioxidant pathways and inhibits the expression of NF-kB, leading to a significant reduction in inflammation [42,47,49,50].

Finally, the ketogenic diet plays an important role in improving several main risk factors for PAD; however, its clinical safety long-term remains unclear.

#### 2.1.4. Intermittent Fasting Diet

Generally, daily food intake is divided into three main meals and two snacks. Intermittent fasting (IF) is a dietary pattern that alternates fasting periods of varying durations with short intervals in which feeding is allowed. Fasting periods often result from an extension of overnight fasting. There are numerous variants of IF or similar modified fasting regimens; however, in general, all these dietary patterns are adequate to provide a correct intake of calories while preserving lean mass, providing the necessary micro- and macronutrients, and ensuring the beneficial effects of fasting with a low dropout rate.

In obese patients, IF is an effective strategy that provides substantial weight loss [51] and leads to a significant change in body composition by consuming fat stores [52,53]. Fasting can influence the composition of the intestinal microbiota by increasing the fermentation of acetate and lactate and activating the thermogenesis of beige cells (due to the upregulation of monocarboxylate transporter 1 in brown adipose tissue), resulting in net fat consumption [54]. One concern is the rapid weight loss due to water loss and glycogen consumption, which may cause fatigue and dizziness [52]. However, growing evidence confirms the remarkable safety profile [55] of IF for the management of overweight and obesity [56,57].

Diabetic patients, particularly those with obesity, showed a noticeable improvement in blood glucose levels during intermittent fasting diets. In fact, a reduction in glycated hemoglobin and an improvement in glycemic control has been documented in these patients regardless of weight loss and body mass transformation [53,55,65,89,570,571,572,573,574,575]. Fasting promotes an optimization of the circadian rhythms and regulation of several molecular and hormonal pathways (such as ghrelin, insulin-like growth factor 1, adiponectin, leptin, 8-isoprostane), which lead to a significant improvement in the function, regeneration and survival of pancreatic β-cells [576,577], with an additional reduction in the risk of developing diabetes [56,578]. IF also modulates the inflammatory state, with a direct effect on insulin sensitivity [58]. Hypoglycemia is a common adverse event in diabetes, but the risk can be significantly reduced with adequate training in the management of hypoglycemic drug therapy during fasting periods [59]. There are few guidelines to support the safe use of an intermittent fasting scheme in diabetes [53]; however, to date, it has proven to be a fairly safe dietary pattern for diabetic patients [60].

In addition to the effects of IF on blood glucose levels and weight management, interesting results were observed in the control of hypertension. In fact, patients with greater adherence to this diet had a better blood pressure profile in parallel with a reduction in abdominal circumference [61,62].

Intermittent fasting can reduce serum cortisol levels [63] and sympathetic tone, as evidenced by a decrease in diastolic blood pressure and heart rate [58]. Furthermore, the decrease in systolic and diastolic pressure during periods of fasting leads to the easier management of hypertension, thus improving cardiovascular risk and quality of life [64]. IF provides significant benefits to cardiometabolic health and positively influences circadian rhythms [65], playing a vital role in the prevention of hypertension and metabolic syndrome [66]. Therefore, a short fasting cycle has shown an effective impact on hypertension and, more generally, on cardiovascular risk [67], with sufficient safety and feasibility [68].

Intermittent fasting can be a therapeutic strategy to achieve better serum lipid profiles and cardiovascular protection [69,70,71] in obese patients at higher CV risk. There is still no solid evidence for the use of IF in clinical practice, however, as most knowledge comes from anecdotal and observational studies or from the experience of patients observing the fasting customs of Ramadan [72,73,579,580,581,582]. Based on the current evidence, IF could ensure a reduction in the serum levels of triglycerides and LDL-c with an increase in HDL-c, with a consequent decrease in cardiovascular risk [72,73]. Part of the benefit derives from the amount of body weight loss and the qualitative change in body mass, with a ripple effect on the respiratory exchange ratio, lipid profile and metabolism, regardless of concomitant physical activity and eating habits [53,74,75,76,77,78,79].

Most studies concerning the impact of fasting on kidney function and its related benefits and safety come from reports of the effects of Ramadan (a pillar of Islamic belief) on patients with CKD [85]. Thus, it is a topic of considerable interest but is equally controversial. Some studies conducted on fasting and CKD document a good safety profile, with a reduction in serum creatinine levels and an increase in the glomerular filtration rate [80,81]. However, several aspects of fasting in CKD concern experts, such as the risk of renal hypoperfusion, which could lead to acute renal injury from dehydration [583], causing hyperkalemia, the worsening of hypertension, metabolic acidosis and proteinuria [584,585]. Attention has now been placed on the safety of fasting in patients with renal insufficiency, suggesting that individuals with mild or moderate chronic renal insufficiency can afford fasting if closely monitored for adequate hydration [82,83] and if the glomerular filtration rate does not appear to deteriorate significantly [84,85]. However, studies are still needed to evaluate the benefits on renal function and assess the safe reproducibility of intermittent fasting in the different phases of CKD to avoid harmful consequences [586].

Atherosclerosis is a chronic low-grade inflammatory disease that involves all arterial beds. The progression of atherosclerotic plaques leading to the narrowing of the arteries relies on endothelial dysfunction and the long-term exposure to oxidative stress. The proinflammatory processes occurring in atherosclerotic plaques appear to be influenced by the metabolic effects of intermittent fasting [77]. The cytokine load promoted by foam cells exposed to oxidized LDL is markedly reduced in patients during intermittent fasting, resulting in the regression of the inflammasome [58,86,87]. The intermittent fasting diet modulates the secretion of various molecules produced by adipocytes. These include adiponectin [88,89], which decreases the expression of adhesion molecules [90,91], preventing the intimal thickening of the vessels and slowing the proliferation and migration of smooth muscle cells [92]; leptin [89], which reduces platelet aggregation and the proliferation of endothelial cells [93]; and resistin [94], which modulates the activity of neutrophils and the endothelial adhesion of monocytes [95]. Fasting reduces liver inflammation and improves glucose and fat metabolism [96]. Fasting also alters the intestinal microbiota, favoring a microbial composition with anti-inflammatory properties, including the modulation of neuroinflammation, local and systemic oxidative stress [56,97,98,587,588,589,590]. Despite an initial increase in macrophage infiltration in adipose tissue and skeletal muscles due to increased lipolysis activity [591], IF promotes weight loss, a reduction in fat mass and the improvement of body composition, with a positive impact on the inflammatory state and related metabolic disorders [58].

The promising results of intermittent fasting in the prevention and treatment of PAD-related risk factors should encourage further studies to evaluate its safe clinical use, long-term effects, indications based on patient characteristics and strategies for minimizing side effects and poor compliance [66,592,593].

### 2.2. Impact of Eating Behaviors on PAD Risk Factors and Predictors

Eating behaviors are a set of habits that an individual possesses or adopts during feeding; they concern the modalities of feeding, the frequency and quantity of meals, the emotional substrate and the influence that food can have on the individual. The main factors that determine our eating habits can include culture, social history, family, individual characteristics and economic and psychological states. The pathological relationship between food consumption and eating behaviors contributes to the development of eating disorders. Some of these disorders contribute to the pathogenesis of atherosclerosis and promote the development of common chronic diseases.

Among the most relevant eating disorders, we selected binge eating, emotional eating, bulimia nervosa, anorexia, sleep disturbances, night eating disorders, home cooking, fast food and the consumption of processed foods and skipping meals for further study. We explored the interesting interaction between eating habits and major predictors of PAD and analyzed how these disorders may be related to the progression of these chronic diseases. The main results are summarized in Table 2.

#### 2.2.1. Emotional Eating, Binge Eating, Bulimia Nervosa, Anorexia Nervosa

Obesity is often the result of several eating disorders, partially due to the impairment of dopaminergic inhibitory/reward neurological pathways [227], emotional dysregulation, the impairment of the hypothalamic pituitary adrenal stress axis [228,229], or the reduced secretion of serotonin secondary to the low intake of tryptophan [230]. Careful assessments of dietary habits should always be conducted, as deleterious eating behaviors often remain underdiagnosed [244]. Binge eating and emotional eating lead to the consumption of large amounts of high-calorie foods in a short period due to a maladaptive response to emotional and psychological distress [231]. The frequency of emotional eating correlates with a higher incidence of weight gain, and in general, the higher the BMI, the more severe the eating disorder. Thus, at the same time, obesity seems to be both the cause and effect of binge and emotional eating [232]. Currently, cognitive, behavioral, psychological and interpersonal therapies along with pharmacological treatments have a beneficial impact on binge control and emotional distress to manage overeating [232].

In recent years, an increasing prevalence of bulimia nervosa in obesity has been observed, describing how obesity is often associated with frequent episodes of loss of control and overeating, which lead to a large consumption of calories [233]. The recurrence of these episodes affects weight gain [234] and is probably related to maladaptive responses to social distress, a tendency toward depression, low self-esteem, performance and body satisfaction [235].

Diabetes mellitus is often characterized by psychological discomfort with regard to nutrition, and an integrated approach for the control of glycemia and the management of eating behaviors is fundamental [594,595]. Dysfunctional eating in diabetes mellitus is associated with a higher risk of acute and chronic complications owing to hyperglycemia and hypoglycemia [596]. Low mood or the recurrence of diabetes-related adverse events could be signs of eating disorders in patients [597].

Patients with DM, especially type 1 DM [236], are often concerned with food, body weight and body shape, with severe psychosocial repercussions, emotive dysfunctions and harmful consequences to health [237,238]. For example, diabulimia is a common condition that leads to binge eating, insulin therapy restriction and self-induced vomiting to control weight or avoid weight gain, leading to dangerous glycemic variability with recurrent episodes of hyper- or hypoglycemia [239]. Binge eating appears to be a risk factor for diabetes [240] and increases the prevalence of depressive symptoms, which complicate glycemic control and lead to worse outcomes [236,241,242].

Impaired hormonal pathways that regulate satiety (e.g., brain sensitivity to incretins) have been observed in diabetes, and this condition is often responsible for several eating disorders such as binge eating, overeating driven by emotional eating and night feeding [241,242,243,244,245]. Among the eating disorders associated with diabetes, anorexia is a rare condition that hampers patient management and increases the incidence of life-threatening complications due to insulin treatment restriction, prolonged fasting and compensatory behaviors (such as the induction of vomiting). This is especially dangerous in patients with type 1 diabetes mellitus, who are more prone to develop ketoacidosis [236]. Controversially, some studies consider anorexia as a protective factor because it leads to lower caloric intake [246]. However, there is no consistent evidence regarding the role of anorexia nervosa in diabetes. In patients with diabetes (especially type 1), the search for maladaptive eating disorders and nutritional compensatory behaviors should be included in the management and prevention of acute and chronic complications of diabetes [598,599,600].

Dyslipidemia is commonly observed in individuals with eating disorders. The lipid profile, characterized by higher serum levels of triglycerides and LDL-c and lower serum HDL-c concentrations, is often the result of deleterious dietary patterns associated with unhealthy eating behaviors. All types of episodes in which a loss of control occurs frequently lead to overeating, with a preference for products with high fat and sugar content that help the patient decompress after the stressful trigger, inevitably worsening the lipid profile even at a young age. Eating disorders associated with an impaired lipid profile and characterized by the loss of control are mainly binge eating and bulimia, which are responsible for an increased risk of developing metabolic syndrome and dyslipidemia [249,250]. The severity of eating disorders appears to directly affect the serum LDL and non-HDL cholesterol levels, leading to an increased risk of cardiovascular events [253]. Furthermore, emotions are strongly linked to metabolic balance, and there is a particular interaction between emotional eating, driven mainly by anxiety, and an increase in LDL-c, confirming that mood plays a significant role in metabolism [252].

Nutrition and eating behaviors contribute to the pathogenesis, progression and management of hypertension because they promote the development of metabolic syndrome or affect complex neuro-metabolic activity. Hypertension is particularly common among individuals with typical uncontrolled overeating episodes, such as binge eating and bulimia nervosa. The emotional and psychological substrate seems to be deeply interconnected with hypertension since individuals who stopped binging showed a consistent reduction in diastolic [247] and systolic [248] blood pressure values. The frequency and extent of binge episodes are predictors of more severe phenotypes of hypertension [249], and individuals with classic binge eating and bulimia nervosa [250] show the highest risk profile [251]. Emotional feeding, particularly in anxiety, has a detrimental effect on the metabolism of hypertensive patients, hampering the success of hypertension management; however, clear evidence on the incidence of hypertension in cases of emotional eating habits is still lacking [252].

The chronic low-grade systemic inflammatory process affecting patients with a loss of control while feeding is further favored by the severity and frequency of binging episodes [248]. Each episode contributes to an increase in serum CRP levels and white blood cell counts, suggesting that eating disorders progressively worsen an individual’s inflammatory profile [254] and increase cardiometabolic risk [248]. Interestingly, elevated levels of inflammatory molecules may be protective against eating disorders, as overactivation of the immune system leads mainly to a lack of appetite. However, emotional, psychological and metabolic involvement during overeating episodes can alter appetite, resulting in dysregulated food intake and the absence of satiety feedback [255].

Several eating disorders exhibit a specific inflammatory profile with a typical composition of cytokines and growth factor production, as observed in binge eating, night feeding and anorexia nervosa, with inter-individual differences in the expression of inflammatory molecules [256]. There is growing evidence showing the fundamental interactions between the gut microbiota, immune system and eating disorders. Binge eating, bulimia nervosa and prolonged fasting in anorexia nervosa have a substantial impact on intestinal dysbiosis by modulating local inflammation [257] and promoting neuroinflammation. The latter alters the brain control of vegetative functions such as satiety, weight and mood [258], contributing to the development and progression of eating disturbances [259]. Several molecular biomarkers have been investigated as possible factors involved in the pathogenesis of inappropriate nutritional habits, including intestinal peptides [257]; endocrine system hormones such as cholecystokinin, ghrelin, glucagon-like peptide 1, YY peptide and pancreatic polypeptide; adipokines such as adiponectin and leptin [260]; neuroinflammatory pathways including phosphorylated NF-kB protein, NF-kBIA and IL-6 gene expression [261]; and cytokines such as IL-1b, TNFalpha, CRP and TGFbeta [258].

#### 2.2.2. Night Eating and Sleep Disorders

Night eating disorder is often associated with obesity [262,263] and weight gain leads to severe sleep disorders [264]. Night eating syndrome (NES) is likely caused by the desynchronization of circadian rhythms with frequent nocturnal awakenings, which are associated with recurrent, large caloric intakes after dinner [265,266]. The complex interplay among nocturnal hyperphagia, lack of sleep and obesity requires further investigation to fully understand this disorder [267,268], and obese patients should always be evaluated for NES to promptly recognize and treat this condition which hampers weight control [269,270]. Additionally, inadequate sleep time and quality can increase uninhibited eating, discomfort and susceptibility to food reward, and impair the endocrine system with the dysregulation of appetite-regulating hormones. This results in hedonic eating rather than a nutritional pattern driven by homeostatic impulses, leading to the development of obesity [271].

Nocturnal food consumption in diabetes affects sleep with a deleterious impact on hormonal and metabolic homeostasis, resulting in a failure to control weight [245] and maintain therapeutic glycemic goals. However, no studies have been designed to correctly evaluate the impact of nocturnal eating on diabetes mellitus. Sleep disturbances are prevalent in diabetic patients, who often suffer from insomnia and obstructive sleep apnea syndrome (OSAS), both associated with the primary development or progression of diabetes [272,273]. In particular, episodes of hypoxia linked to OSAS and the neurohormonal imbalance due to the short duration of sleep seem to be one of the complex causes of impaired metabolic homeostasis and lack of glycemic control [274,275,276,277]. Diabetic patients with OSAS treated with continuous positive airway pressure (CPAP) demonstrated a significant improvement in glycemic control and insulin sensitivity, parallel to the reduction in snoring and daytime drowsiness [278]. Sleep duration modulates satiety by influencing metabolic and neurological pathways. Insufficient sleep time promotes appetite by increasing serum leptin levels and lowering ghrelin production [279] with a reduction in satiety. It also alters the perception of total food consumption, resulting in excessive caloric intake, obesity and diabetes mellitus [280]. In addition, sleep disturbances include late-night caloric intake, a condition characterized by an excessive delay in dinner time or by the frequent consumption of night-time snacks with a significant increase in the risk of diabetes and related complications [281]. Therefore, sleep quality and duration should be included in clinical assessments to identify patients at the highest risk of diabetes and those failing to achieve glycemic goals [282,283], in the young population [284,285] and in type 1 DM [280,286,287].

Sleep habits change the lipid profile by affecting metabolism with a U-shaped association [290,291,292]. Seven hours is the ideal mean sleep duration, which has been observed to be linked to the greatest benefit in preventing metabolic syndrome, lipid dysmetabolism and mortality [293,294,295]. Sleep disturbances induce a decrease in serum concentrations of leptin, resistin and HDL-c, favoring an increase in the levels of hepatic cholestatic indices and an accumulation of hepatic cholesterol due to the under-expression of CYP7A1, which is responsible for the conversion of cholesterol into bile acids [296]. Among sleep disorders, obstructive sleep apnea has the most significant impact on dyslipidemia [297]. Hypoxia caused by proximal airway obstruction triggers the release of fatty acids from fat stores and the liver as part of the effects of adrenergic pathway activation, resulting in the higher synthesis of triglycerides and cholesterol esters. Furthermore, neuro-adrenocortical hyperactivation [298], triggered by hypoxic episodes, favors the synthesis, accumulation and oxidation of LDL cholesterol due to lipoprotein lipase dysfunction. The treatment of OSAS, with the loss of weight and continuous positive airway pressure (CPAP), reverses alterations in the lipid profile [299]. Patients with sleep disorders often tend to eat late at night or eat after-dinner snacks. This deleterious habit accelerates the progression of atherosclerosis, with the premature degeneration of the arterial wall and increase in arterial stiffness [601]. The irregular consumption of food at night reduces the time to sleep and disrupts the physiological overnight fast with excessive serum lipid loads after dinner, leading to severe impairments in post-prandial lipid metabolism [300,301]. At night, there is a physiological reduction in metabolic activity and consequently poor tolerance to caloric intake [602], as demonstrated by the harmful blood peaks of triglycerides, fatty acids and cholesterol esters during night feeding [302].

Short sleep duration runs in parallel with poor quality diets, characterized by a large intake of high-calorie foods rather than an adequate intake of vegetables and fruit. Hypertension might be the consequence of this complex interplay between sleep disturbances and metabolism, with variability in the blood pressure profile according to the impairment of the food reward pathway, imbalance of leptin and ghrelin production and circadian rhythm integrity. Frequent conscious and unconscious interruptions in night rest are responsible for abnormal nocturnal blood pressure profiles [288]. Furthermore, emotional involvement resulting from sleep disturbance frequently leads to relieving discomfort by eating [603]. Night eating has a strong impact on blood pressure and arterial stiffness by compromising the physiological metabolic cycles [289], which causes unwanted weight gain due to increased total daily caloric intake and is typically associated with a low consumption of plant-based foods [601,604].

#### 2.2.3. Skipping Meals

Social and environmental factors often lead to poor adherence to the typical distribution of meals (breakfast, lunch, dinner and two snacks). Irregular eating patterns with frequent skipping of meals and/or snacks are widespread in the population, especially among young people [605,606,607]. In particular, skipping breakfast leads to an increase in calorie intake during the rest of the day, resulting in an unbalanced distribution of food that favors the development of obesity [303,304,305].

Eating within certain time intervals plays a fundamental role in glycemic control as the regular intake of food improves glycated hemoglobin levels and reduces the incidence of diabetes in younger age groups. In addition, not eating one or more meals throughout the day is a practice often pursued by people to lose weight, but the main result is an increased risk of developing diabetes [306]. The frequent and random skipping of meals worsens diabetes control by altering glucose tolerance. On the other hand, a well-established distribution of daily calories results in an effective dietary strategy that counteracts the harmful effects of skipping meals, although it is essential to avoid the excessive glycemic fluctuations caused by multiple meals [307]. Breakfast appears to be the most relevant meal because it is responsible for energy balance, insulin secretion and sensitivity, glucose metabolism, fat oxidation and post-prandial inflammatory responses [308]. Skipping meals, especially breakfast [305,309,310,311], exposes diabetic patients to severe glycemic variations [312], long-term vascular damage and a higher incidence of MACE and MALE [313]. The effects of skipping breakfast on glycemic control can also be observed in young people who increasingly have incorrect eating behaviors related to a higher incidence of metabolic syndrome and susceptibility to diabetes, as well as in pregnant women with an increased risk of developing gestational diabetes [314]. Curiously, these effects can also be observed in the very short term even after skipping a single breakfast, with higher glycemic values after lunch [315]. As expected, skipping meals is often associated with other deleterious eating habits, such as frequent fast food consumption, late night eating, stress [316] and overeating, conditions that lead to the development of diabetes [608].

Eating at home is associated with a reduced habit of skipping meals, which has also been shown to contribute to dyslipidemia. Breakfast is usually the most skipped meal with the most relevant effects on the lipid profile [305], and this trend is often combined with other unhealthy eating habits, such as the frequent consumption of pre-cooked and packaged foods and poor home cooking skills. In some situations, the intentional avoidance of certain meals is driven by the desire to lose weight. However, skipping meals with an irregular pattern is an unsafe and ineffective long-term strategy for weight control and leads to early metabolic alterations such as an increase in serum LDL-c levels [304,317]. Skipping meals makes metabolism less efficient, especially in the case of breakfast which prepares the body to digest and metabolize food with a better post-prandial lipid profile [308].

Irregular fasting also plays a role in the progression of metabolic factors and biological patterns underlying hypertension. Frequent meal skipping leads to a higher rate of failure in blood pressure control. Indeed, evidence confirms the negative effect of skipping meals, especially breakfast, on hypertension and metabolism [305].

The irregular distribution of calories during the day has become common in both younger and older generations and may promote immune system dysregulation [609], leading to low-grade intestinal inflammation and dysbiosis with harmful health consequences [308]. Eating disorders and detrimental eating habits are risk factors that should be considered when identifying patients at higher risk for cardiovascular complications and who deserve targeted therapy to control their food disturbances [610].

#### 2.2.4. Home Cooking, Fast Food Access, Ultra-Processed and Packaged Food Consumption

Another cost-effective preventive strategy for obesity is to increase individual home cooking skills [318,319] from a young age [320], favoring healthy cooking methods such as roasting, sprouting and boiling over fried foods, maximizing the anti-obesity properties of some ingredients rich in phytochemicals (e.g., soy and fiber) [321]. Home cooking impacts obesity by improving individual skills in the selection of ingredients [322] and reducing the uninhibited and unconscious consumption of calories [323]. Furthermore, home cooking minimizes access to fast food, allowing more effective weight control [324,325] and a healthier dietary intake, including a lower consumption of fried meals, salt, sugary drinks, processed foods and saturated fats contained in low-quality oils, in parallel with a greater intake of fruit and vegetables [324,326,327,328]. Home cooking reduces the consumption of low-quality packaged meals, characterized by high caloric density and additives [329], which often lead to the development of obesity and endocrine disorders [330] such as higher serum estrogen levels due to polycarbonates and epoxy resins in plastic materials [331].

People who eat homemade meals and improve their home cooking skills show a greater ability to select quality foods and a greater understanding of the impact of nutrition on health, with a lower risk of developing diabetes [334]. The consumption of home-cooked meals is a reliable predictor of increased adherence to healthy eating habits, and home-cooking education programs are an effective diabetes prevention strategy [335].

Food quality is defined by the type of cooking used to prepare a meal, and this can be associated with the incidence of diabetes. In particular, stewing or boiling meat reduces the risk of developing diabetes [336] compared to cooking meat at a high temperature (e.g., barbecuing, roasting, grilling). These high-temperature methods release harmful compounds, including advanced glycation end products (AGE) and heterocyclic aromatic amines which lead to inflammation and glycemic toxicity [337]. Oil is a key ingredient in food preparation and the use of extra virgin olive or soybean oil (compared to animal fat, peanut oil, refined vegetable oil, rapeseed oil and sesame oil) provides consistent protection against diabetes [338].

The impact of cooking plays a major role in all rural and low-income countries, where cooking patterns are driven by sociocultural traditions and where the perception of the impact of food and diet on health is often low. Globalization has invaded the markets of these countries, providing a cheaper but harmful alternative to healthy food [611,612]. Fast food addiction has resulted in a pandemic distribution affecting people of different ages, cultures and origins [613]. Fast food is an inexpensive, highly palatable and addictive alternative to home cooking. However, the low quality of the ingredients, the large portions of the meals and the excessive intake of sugars, salt and calories are all characteristic traits of this harmful dietary pattern strongly associated with insulin resistance and diabetes [339,340,341,342]. The main ingredients of fast-food meals are processed red meat (e.g., sausages, hamburgers), fried products (e.g., French fries or fried chicken) and other foods rich in added and refined sugars (e.g., packaged ice cream and sweetened beverages), high glycemic index products, saturated fats and salt, all of which contribute to the development of diabetes [343,344]. The long-term consumption of fast food is associated with hepatic steatosis. Excessive liver triglyceride alters lipid metabolism, increasing post-prandial serum lipid levels and reducing insulin sensitivity [345,346]. Furthermore, steatosis reduces the liver’s ability to process cholesterol and saturated fatty acids contained in large quantities in many fast-food products and the consequent lipid overload impairs pancreatic beta-cell function [347].

Home cooking increases the consumption of vegetables and fruits rich in phytochemicals that lower LDL-c [321]. The consumption of ultra-processed and packaged foods is one of the most important risk factors for metabolic syndrome, contributing to the aggravation of dyslipidemia [357]. Ultra-processed foods are characterized by low-quality and high-calorie density products; this diet comprises a substantial part of global nutrition due to the addictive taste, low cost and wide availability of these foods. In parallel with the increase in the incidence of metabolic syndrome, dyslipidemia and other cardiovascular risk factors, fast food and ultra-processed foods have become more prevalent in diets globally and are frequently associated with other unhealthy eating behaviors [358,359]. In recent years, younger generations [360,361] have been increasingly exposed to junk food such as processed red meats, fried foods and high-calorie meals rich in refined sugars and salt. This may explain the current worsening of individual nutritional conditions (such as obesity, diabetes and dyslipidemia [362]), the early incidence of chronic diseases and higher morbidity.

Hypertensive patients should consider improving their cooking skills, as a reduction in blood pressure has been documented in individuals who have greater awareness of the ingredients used when cooking. In particular, olive, soy and peanut oils used in cooking have beneficial effects on endothelial function and play an important role in the management of blood pressure [348,349]. Patient education in the selection of ingredients, in the correct quantity and in the preparation of food has resulted in a significant reduction in the incidence of adverse cardiovascular complications through the better control of hypertension [350]. Cooking educational programs and home food preparation guidelines should be included in an active therapeutic approach for blood pressure management. Moreover, minimizing access to restaurants or the intake of processed foods [351,352] are key strategies to counteract the incidence of chronic comorbidities that affect hypertension [353] and related outcomes [354] in both young and adult patients [355,356]. Although home cooking training reduces junk food consumption [356,614], patients are rarely educated in proper meal preparation [348,349].

To the best of our knowledge, no recent studies have assessed the impact of eating disorders on the incidence and progression of kidney disease. There are data on cooking techniques as a possible protective factor against a common complication of renal failure, namely hyperkalemia. For patients with renal insufficiency at risk for hyperkalemia due to the consumption of potassium-rich foods, boiling should be the preferred method of cooking meals, as it correlates with a greater reduction in potassium content [363]. Consuming frozen vegetables [364], soaking and double-cooking foods are additional dietary skills that can help minimize potassium content [367]. Low-quality processed foods and packaged meals are excessively high in potassium, phosphorus and protein, as well as calories [368,369,370].

Patients with chronic kidney disease and poor nutrition education often show a higher frequency of fast food and junk food intake [365], along with restricted access to fresh produce and high-quality food. These harmful eating behaviors and a suboptimal diet could worsen renal function in this group of patients, increasing the risk of cardiovascular complications [366].

There is little awareness of the role of cooking habits in the promotion of inflammation; however, the wise selection of foods and ingredients could be an effective strategy for reducing the inflammatory burden. Moderate amounts of red meat and processed foods or even a full transition to other protein sources (poultry, soy, legumes) can reduce the oxidative stress derived from food metabolism [371,372,373] regardless of weight loss and calorie intake [374].

Ultra-processing is the extraction of substances such as fats, starches and sugars from foods and adding as artificial colors, emulsifiers, flavorings and stabilizers that mimic the sensory qualities of natural foods while affecting the homeostasis of the intestinal microbiota. Such ultra-processed foods worsen serum lipid levels, induce the release of proinflammatory molecules from adipose tissue [375] and reduce the antioxidant defenses [376], resulting in cardiometabolic risk [615,616,617]. Unfortunately, these products are affordable, highly palatable and easy to find, facilitating their prevalence in patient diets and accelerating the incidence of atherosclerotic disease in recent years [618].

### 2.3. Smoking and Eating Behaviors

Smoking is a well-established cardiovascular risk factor deserving a separate discussion as very little is known about its relationship with diet, eating behaviors and food disorders. A peculiar coexistence of unhealthy lifestyle habits, smoking and traditional risk factors for atherosclerosis, such as obesity, physical inactivity and diabetes, is frequently observed in patients with a high CV risk.

Many concerns arise from the prevalence of smoking in younger generations, who are also those most exposed to improper diets and lifestyles [619]. The concomitant presence of several dietary risk factors is common in smokers, including lack of sleep, night eating [268], skipping meals [377], emotional eating [378], low-quality diet [379,380] and the low consumption of foods of plant origin, often replaced by fast food and ultra-processed foods [381]. Nutritional programs should run in parallel with the smoking cessation process; notably, these programs can prevent the excessive caloric intake [382] that often accompanies withdrawal symptoms [383]. Women who quit smoking are particularly prone to relapse in proportion to their degree of concern about physical appearance, depressive mood and weight gain [620]. An adequate educational program with healthy, non-addictive and anti-inflammatory diets [621] could counteract the oxidative stress caused by tobacco and could correct eating disorders related to smoking cessation. The nutrition needs of smokers should be considered to ensure the effective prevention of the cardiometabolic risk to which they are exposed. Further studies are needed to explore the possible etiology and molecular pathways underlying the interaction between smoking, food and habits.

## 3. PAD of Lower Limbs and Nutrition

### 3.1. PAD and Nutritional Status: Obesity and Malnutrition

Addressing the growing incidence and prevalence of CV disease has become a priority to save millions of people worldwide [622] and to reduce its economic burden [623]. Primary prevention plays a pivotal role in reducing the disastrous consequences of atherosclerotic disease. In particular, prevention based on an adequate nutritional program could lead to extraordinary results for coronary heart disease, cerebrovascular disease and PAD [402].

Patients with PAD of the lower limbs are burdened by a severe atherosclerotic phenotype characterized by polyvascular involvement that makes these individuals particularly fragile and at a high risk for disabling complications [624]. It is necessary to develop a well-structured nutritional program and investigate the presence of eating disorders to stratify risk and act promptly to prevent detrimental complications.

Obesity in PAD patients is a common clinical feature that has a significant impact on comorbidities and outcomes [467,625]. Indeed, weight gain in patients with LEAD is associated with more severe clinical scenarios [626] such as a reduction in walking autonomy, fast progression towards critical ischemia and an increased probability of failed revascularization treatments [386,471,627,628]. Furthermore, an obese patient with PAD has a more aggressive phenotype of concomitant comorbidities, which hinders the achievement of recommended therapeutic goals and results in lower survival [629,630]. Finally, obesity in PAD is a reliable predictor of negative outcomes, leading to a higher incidence of MACE and MALE. Indeed, obese patients with LEAD have an approximately 1.5-fold increase in the development of critical limb-threatening ischemia (CLTI) regardless of other confounding factors [436]. Therefore, weight loss is an effective and safe approach for the simultaneous management of CV risk factors, comorbidities and PAD, with a significant gain in terms of survival and quality of life [628,631]. A high-calorie diet that does not guarantee an adequate supply of proteins, vitamins and minerals could lead to nutritional deficiencies despite obesity, and many obese patients with PAD of the lower limbs suffer from sarcopenic obesity and/or selective malnutrition, which further worsens nutritional status [402,632,633]. Patients often have calorie–protein malnutrition, which is responsible for the harmful condition of cachectic sarcopenia. This increases the risk of adverse events and accelerates atherosclerosis by favoring a more aggressive inflammatory state [634,635,636]. In patients with advanced stages of PAD, the diet should provide sufficient calories and protein to match the increased resting energy expenditure and intense metabolic stress secondary to ulcer healing and chronic infections [394]. Interestingly, malnutrition is a dire condition in which obesity in PAD paradoxically appears to be a protective factor compared to cachectic sarcopenia, as obese patients exhibit fewer complications and longer survival than lean individuals. This obesity paradox was confirmed regardless of age, sex and comorbidity [437]. In fact, sarcopenia carries an independent additional risk of MACE and MALE. In particular, patients with advanced stages of lower extremity arterial disease (LEAD) who undergo a revascularization have dramatic short-term outcomes with increased rates of MACE [438,439] and major amputations [440], slightly improved by the adherence to the best medical treatment such as aspirin and statin [441]. Therefore, sarcopenia may be an aspect to consider for the risk stratification of patients with CLTI in terms of the success rate of limb salvage [442].

The nutritional status of patients with PAD plays an important role in their outcomes [637]. An integrated assessment of individual nutritional status would improve the residual CV risk through the integrated management of diet and eating habits [638].

### 3.2. PAD of Lower Limbs and Nutrients

The inflammatory process underlying atherosclerosis in PAD is responsible for endothelial dysfunction and triggers vascular inflammation and thrombosis, leading to the progressive narrowing of the vessels in the lower limbs. Diet modulates the vascular inflammatory process and slows atherosclerosis [402]. The minimal consumption of red meat, processed meat and saturated fats should be the first nutritional aspect to be corrected to prevent or even significantly alter the progression of PAD. In addition, typical products of the Western diet, which are linked with a greater risk of LEAD, should be avoided, and a greater intake of plant-based foods should be recommended for this population. Phytosterols (derived from nuts and vegetable oils) and flavanols (mainly present in fruits and vegetables) play a fundamental role in the lipid profile and inflammatory process by reducing LDL-c, the formation of foam cells, the oxidation of ectopic cholesterol deposits on arterial walls and the endothelial production of cytokines and chemokines [639,640,641,642,643].

A dietary or supplemental intake of vitamins has been documented to have additional beneficial effects on atherosclerosis in PAD [637,644]. Vitamins with an anti-inflammatory function (especially vitamins A, C and E) are excellent stabilizers of atherosclerotic plaque by modulating local oxidative stress, resulting in decreased plaque vulnerability and risk of rupture [394,645,646,647,648,649,650,651]. Furthermore, B vitamins (B2, B6, B9 and B12) reduce serum levels of homocysteine, which promotes arterial stiffness by increasing collagen production [652,653], induces the proliferation of vascular smooth muscle cells and facilitates local inflammation and thrombosis [654,655,656,657]. Vitamin deficiency should be investigated in patients with advanced stages of LEAD as this may accelerate progression to CLTI. Indeed, lower plasma levels of vitamin B12 were found in patients with DM and LEAD who underwent a major amputation, leading to worse outcomes during post-surgical rehabilitation [443]. The supplementation of activated vitamin D to lower extremity PAD patients on dialysis may be a cost-effective strategy to reduce the risks of foot infection and MACE [444] and the rate of major amputations [445]. Although the role of vitamin C in non-healing foot wounds is unclear, given its relative safety, dietary supplementation could be another strategy to reduce lower extremity complications [446,447].

Molecules with antioxidative properties such as zinc and coenzyme Q exert a beneficial effect on the management of hypertension, dyslipidemia and diabetes in patients affected by PAD of the lower limbs [402]. However, the impact of micronutrients (such as zinc and magnesium) on foot ulcer healing is uncertain and there is no evidence on their impact on the prevention of CLTI progression, amputation rate or failure of revascularization. Therefore, supplementation is not routinely recommended in clinical practice [448].

Dietary fats are vital as they are a rich source of energy, contribute to the synthesis of hormones and take a structural part in the composition of cell membranes. However, the excessive intake of saturated fat, cholesterol and triglycerides promotes inflammation and accelerates the formation of atherosclerotic plaques. Therefore, therapeutic diets should include polyunsaturated fatty acids (PUFAs), which have anti-inflammatory and anti-thrombotic effects, regulate blood pressure and improve claudication symptoms by increasing nitric oxide production [637]. Humans do not have the ability to synthesize PUFAs, which are mainly present in fish and vegetable oils, nuts and seeds and foods with known anti-atherogenic effects [658,659]. Individuals with advanced stages of PAD undergoing revascularization may require supplementation of their diet with PUFA-rich foods given that low plasma levels of eicosapentaenoic acid correlate with the incidence of MALE and MACE [449,450].

Soluble fiber has remarkable health benefits in patients with PAD [660]. Each meal should include an adequate intake of fiber as it significantly impacts post-prandial hyperglycemia and hyperlipidemia by slowing stomach emptying and the intestinal absorption of dietary fat and glucose, resulting in lower glycemic/lipemic variability [661]. Soluble fiber exerts a beneficial effect on weight, blood pressure and lipid profiles through several mechanisms. The sense of satiety following fiber intake leads to weight loss and a reduction in the release of inflammatory and vasoactive molecules from fat deposits. Vegetable fibers also modulate the excretion of bile acids and, consequently, decrease serum cholesterol levels. Finally, soluble fibers modulate the intestinal microbiota, causing an increase in the production of short-chain fatty acids with beneficial effects on inflammation and metabolism [662]. Fenofibrate is a synthetic derivative that converts to fibric acid, which directly modulates PPAR-α activity, and a similar effect is observed for soluble dietary fibers [663]. The Fenofibrate Intervention and Event Lowering in Diabetes (FIELD) [664] and ACCORD [665,666] trials demonstrated that fenofibrate in diabetic patients reduces the risk of microvascular complications, including lower extremity amputation (LEA) [451,452,453,454,455] even after the aggressive management of blood pressure, glycemia and cholesterol [667].

Beneficial effects on lower extremity outcomes seem to correlate with the modulation of the adiponectin-dependent pathway [668] and the activation of PPAR-α, a transcription factor involved in lipid metabolism, ketogenesis and peroxisomal/mitochondrial fatty acid β-oxidation [669]. Interestingly, patients with a consistent intake of fiber often manifest other beneficial habits and healthy behaviors [670].

### 3.3. PAD of Lower Limbs and Diets

#### 3.3.1. Mediterranean Diet

The MD embodies many of the suggested nutritional recommendations for patients with PAD. It is rich in polyunsaturated fats due to the use of olive oil, nuts and seeds, which represent the main source of dietary fats rather than dairy products. Moreover, the intake of fibers and other vegetables is included in almost all meals, while red and processed meats are mainly replaced by fish, lean meat and vegetable proteins [402].

This diet plays an important role in both the primary and secondary prevention of PAD [671], as a lower incidence of obliterating the arteriopathies of the lower limbs in populations at risk has been documented [403,404]. The progression to symptomatic claudication is reduced with strict adherence to the MD [405]. Finally, in patients with LEAD, the adoption of the MD may offer significant advantages in terms of lower mortality and morbidity, with protective effects on MACE and MALE [398]. Therefore, the Mediterranean diet appears to be an extremely effective preventive and therapeutic strategy to be recommended in PAD.

This dietary model exerts its beneficial effects by counteracting the inflammatory process connected to atherosclerosis and improving the management of other risk factors and comorbidities affecting patients, such as glucose levels in diabetes, blood pressure values in hypertension and weight loss in obesity [403]. However, the main limitation that hampers compliance with this diet is its high cost. Indeed, a higher level of education and wealth is correlated with a higher level of adherence to the MD [672] and a reduction in common risk factors associated with a suboptimal diet [673]. The consequences of globalization and expected income growth may reduce socioeconomic disparities and facilitate access to healthy food [674] even in low-income countries [538]. Unfortunately, the population affected by PAD is frequently characterized by more severe social limitations and economic difficulties, which are closely associated with a higher incidence of vascular complications including MALE [457]. Indeed, in some countries, socioeconomic disparities that vary among ZIP codes associated with lower-income areas have a higher incidence of amputations [456].

#### 3.3.2. The Vegetarian, Vegan and Plant-Based Diets

Plant-based eating patterns limit the intake of animal proteins, replacing them with fruits, vegetables, grains, nuts, seeds and legumes [394]. Due to the restriction of meat, dairy and all other animal products, the vegan diet has been widely espoused as an effective anti-atherogenic, non-drug treatment. In fact, the incidence of cardiovascular diseases in the vegan/vegetarian population is significantly reduced compared to those who follow a Western-style diet, and the lower consumption of saturated fats, salt, sweeteners, cholesterol and animal proteins decreases the risk of diabetes mellitus, hypertension and dyslipidemia. Patients with PAD have a high atherosclerotic burden and are affected by various comorbidities that facilitate disease progression and the incidence of adverse events. Adherence to a plant-based diet has shown promising results in the PAD population, with direct benefits on patient mortality and morbidity, especially in individuals at greater risk such as smokers [391]. The vegetarian diets play a cardioprotective role in PAD of the lower limbs by limiting vascular damage and partially reversing atherosclerotic plaque formation. Foods of plant origin also improve endothelial function, prevent the ectopic accumulation of LDL and oxidation on the arterial walls and mitigate vascular inflammation by counteracting the formation of foam cells with an overall reduction in atherosclerotic complications [392]. The large intake of phenolic compounds and PUFA in this diet counteracts the incidence of PAD and reduces the occurrence of MACE by improving lipid metabolism [392] and facilitating the achievement of the LDL-c recommended targets [393], which are a primary goal to prevent disabling complications and death in high-risk populations [394,395]. When meat and other animal products are mainly replaced by foods of plant origin, the production of trimethylamine-N-oxide (TMAO), a metabolite of the intestinal microbiota that contributes to atherosclerosis in PAD, is reduced [396]. Higher levels of TMAO have been observed in CLTI and sustain the atherosclerotic process underlying vessel narrowing and systemic inflammation [397,398,399,400]. The modulation of the intestinal microbiome and its metabolites through the adoption of vegetarian dietary schemes is a convincing strategy to slow down atherosclerosis, and the beneficial effects of plant-based foods provide a further contribution to the control of all risk factors and comorbidities for PAD [401].

#### 3.3.3. Low Carbohydrate Ketogenic Diet

A ketogenic diet provides a very low daily intake of carbohydrates (less than 50–30 g/day), inducing ketone production to meet the energy needs of the nervous system with a fairly safe profile in short and controlled periods [675]. No dedicated studies have been published on the ketogenic diet in a population with PAD; however, the nutritional benefits in CV disease have been extensively evaluated. LEAD results from comorbidities and uncontrolled risk factors that can be partially reversed or at least improved with short cycles of the ketogenic diet.

Diets with a very low carbohydrate content and adequate caloric intake from proteins and polyunsaturated fats allow significant weight loss by counteracting the effects of obesity on blood pressure, the lipid profile and insulin resistance. Thus, improvements in the management of dyslipidemia, diabetes mellitus and hypertension in atherosclerotic diseases can be achieved by adopting this dietary model with a reduction in the overall CV risk [384]. Furthermore, ketones appear to exert a direct effect on inflammation and endothelial function, with a consequent reduction in cardiovascular adverse events, modulation of the inflammatory state and maintenance of vascular homeostasis [385].

The ketogenic diet in LEAD could be alternated with a Mediterranean dietary model for short periods to facilitate weight loss and improve the control of CV risk factors in selected patients with preserved renal function and high compliance with carbohydrate restriction [44].

#### 3.3.4. Intermittent Fasting Diet

Overeating and excessive caloric intake are important causes of obesity in PAD, a condition that inevitably worsens the expression of concomitant risk factors, resulting in a more aggressive disease phenotype. IF promotes significant weight loss by limiting feeding time and prolonging overnight fasting, with a significant beneficial effect on PAD and the comorbidities associated with this condition [386].

The beneficial effect of IF on CVD was first believed to be exclusively “weight-focused”, and doubts arose about the safety and efficacy of IF in the comorbid patient regardless of the weight loss achieved [387,388]. Recently, a central role of IF on the modulation of the inflammatory process was confirmed. IF significantly reduces oxidative metabolic stress, modulates the intestinal microbiota and regulates the activity of the immune system, contributing to the slowdown of the atherosclerotic process [77].

IF demonstrated consistent results in stabilizing atherosclerotic plaques in peripheral arterial beds by influencing the composition of the lipid core. Furthermore, IF appears to modulate the endothelial expression of proinflammatory molecules that trigger the local activation of macrophages and lymphocytes, reducing the vulnerability of plaques.

Fasting regulates liver function by decreasing the hepatic accumulation of triglycerides, which results in an improvement in lipid metabolism and a decrease in circulating LDL-c levels, thereby preventing the progressive ectopic deposition of oxidized LDL molecules on the arterial wall [389]. IF could be a strategic approach for all patients with PAD not at risk of hypoglycemia and osteoporosis [390], obesity and severe inflammatory burden; however, further studies are needed to evaluate the possible implications of IF in PAD [77].

### 3.4. PAD and Eating Behaviors

An assessment of the possible interaction between eating behaviors and PAD is lacking due to the need for the comprehensive management and risk stratification of patients. The role of food disorders in the incidence and progression of risk factors underlying PAD of the lower limbs has been extensively explored; however, no large, dedicated evidence has been published on the nutritional habits of patients with PAD.

Emotional eating, binge eating and bulimia are characterized by impulsive eating, which is associated with a higher risk of cardiovascular complications by accelerating the atherosclerotic process [406]. Eating disorders, which include overeating due to a loss of control, are psychiatric disorders with the highest mortality rate due to MACE and other disabling adverse events related to atherosclerosis [407]. Moreover, patients with CLTI with extensive tissue loss due to diabetic/ischemic foot injury or those who have undergone amputation often experience an emotional burden due to progressive functional disability and physical transformation. Eating patterns are influenced by the individual emotional sphere, often resulting in a broad spectrum of eating disorders, such as emotional eating and binge eating, with a further worsening of the clinical conditions and outcomes of patients with LEAD [458]. An evaluation of the eating behaviors of patients could help identify those habits to correct, contributing to a more effective management of all the risk factors associated with PAD.

Indeed, the correction of nutritional habits and the treatment of eating disorders have shown significant reductions in blood pressure, obesity and metabolic deterioration [247]. It is interesting to note that adipokines [410], peptides produced by adipocytes with regulatory functions in metabolism, are markedly influenced by binges and psychological overeating [408]. Various adipokines (such as adiponectin, omentin-1 and sortilin) play important roles in the vascular complications of diabetes [409] and in the incidence and progression of PAD. Furthermore, serum adipokine levels can help predict outcomes after a revascularization procedure [410,411]. Thus, the evaluation of eating behaviors might be a promising predictor of MACE and MALE in the population with PAD [412,413]. Sarcopenia is a common finding in arterial disease of the lower extremities, and the prevalence of anorexia and cachexia is high, especially in the advanced stages of PAD.

Malnutrition driven by eating disorders leads to disastrous consequences such as arterial oxidative stress, systemic inflammation and muscle trophism, often resulting in the failure of available therapeutic strategies such as revascularization [414].

Night sleep has been evaluated as a novel predictor of cardiovascular disease. Sleep duration and quality can influence the severity of PAD risk factors, affecting patient mortality and morbidity. However, it is interesting to note that abnormal sleep can directly provoke peripheral artery endothelial dysfunction and stiffness, which are known subclinical markers of atherosclerosis and early PAD [415].

Smoking and obesity are the main causes of obstructive sleep apnea (OSA) and are the principal risk factors for patients with PAD of the lower limbs. However, OSA is still underdiagnosed and underestimated in patients with PAD of the lower limbs [416,417,418,419,420]. OSA accelerates the progression of comorbidities in patients with PAD [421] and worsens atherosclerotic burden. The severity of OSA is linearly associated with the growth and vulnerability of atherosclerotic plaques [422] and, consequently, with more advanced stages of PAD, including CLTI [423]. Therefore, sleep disturbances in patients with LEAD should be promptly identified and treated to reduce the incidence of MACE and MALE that lead to poor outcomes [424,425], including amputation [459].

Glycemic variability, defined by glycemic excursions, has recently been studied as a new parameter of glucose metabolism and as a therapeutic target to control HbA1c. It accelerates atherosclerosis, alters endothelial homeostasis, and contributes to the progression of PAD and CAD in patients [676].

Skipping meals is a common harmful eating behavior mostly prevalent in younger generations. Although fasting leads to increased fat oxidation with initial weight loss, the long-term adoption of this habit is associated with metabolic inflexibility that impairs serum glucose control and increases oxidative stress [308,315]. Skipping meals, especially breakfast, worsens post-prandial lipid [426,427,428] and glucose levels, which increases the severity of glycemic excursions, resulting in greater exposure to glycemic variability toxicity [429].

Notably, glucose variability plays an important role in endothelial dysfunction and arterial stiffness, which are early steps in atherosclerotic, leading to Peripheral Arterial Disease of the lower limbs [430]. Hyperglycemia correlates with worse outcomes in PAD [431], and post-prandial glycemic variability due to skipping meals contributes to metabolic deterioration and increases their morbidity and MACE rates [432,433].

Fast food products and ultra-processed foods have become commonplace in many diets, and the increase in the consumption of industrially manufactured foods parallels the current pandemic of obesity, hypertension, dyslipidemia, atherosclerosis, diabetes mellitus [677,678] and major cardiovascular complications such as Peripheral Artery Disease [434].

Home cooking could minimize access to processed foods and improve individual skills in the selection of ingredients with beneficial properties that counteract the deleterious effects of incorrect eating behaviors commonly found in the population with LEAD [435]. Special awareness should be reserved for the eating behaviors of patients with advanced stages of PAD who experience limitations due to amputation or tissue loss and lack of social support. Physical barriers to food supply and cooking are often a major factor in suboptimal cooking habits resulting in the increased consumption of more affordable yet low-quality food with the further deterioration of patient outcomes [460].

## 4. Discussion

Millions suffer from the disabling consequences of atherosclerosis yearly, making it the leading cause of morbidity and mortality worldwide. PAD represents the atherosclerotic involvement of the arterial beds excluding the heart and brain. The progressive narrowing responsible for reduced blood flow to peripheral tissues ultimately results in severe ischemic damage [679,680].

Atherosclerotic vascular complications have a global pandemic distribution along with a dramatic increase in the prevalence of traditional risk factors associated with the development and progression of PAD. Regardless of the revascularization strategy and the stage of the disease, the management and correction of traditional risk factors related to PAD of the lower limbs represents a fundamental approach that should be pursued in all patients to reduce the deleterious consequences of this disease [508].

However, the definition of “traditional” may not include many other underestimated risk factors that have shown a relevant impact on the progression of PAD and the incidence of major complications such as MACE and MALE. Dietary risk factors belong to this group of lesser-regarded predictors of PAD. However, while current guidelines recognize the value of nutrition as the primary and secondary prevention of arterial disease from a young age or even in the peripartum phase [681], the awareness of its impact on health by both patients and physicians remains below expectations [1].

Dietary risk factors, including unbalanced diets, low-quality foods and suboptimal eating behaviors, are responsible for the onset and progression of classical risk factors, ultimately leading to the development of PAD [405]. Indeed, scientific evidence on the role of nutrition in atherosclerosis and typical vascular complications does not reflect the urgent need to expand our knowledge of this extremely important aspect of medical management. The assessment of common eating habits and possible disorders prevalent in PAD should be the focus of future research.

Current guidelines provide general dietary recommendations for the management of atherosclerotic disease but do not allow for the personalization of an effective dietary approach based on a wide variety of patients and their characteristics. Each diet exerts a specific beneficial effect by modulating the quantity, quality and methods of food intake. Through a consistent understanding of the benefits derived from the diet and through mastery of the therapeutic potential of nutrition, we could customize a dietary intervention according to the needs of the individual [682].

Individuals with PAD exhibit different stages and clinical presentations of the disease based on the individual expression of genetic and acquired factors, including the severity of each comorbidity and known risk factors [683]. Therefore, a future goal might be to select dietary models that can be adapted to the frailties, characteristics and goals of each person to apply the therapeutic properties of each diet to meet the needs of each clinical scenario.

Furthermore, treatments based on nutritional interventions should not be limited to simple recommendations as to which foods are to be included in the diet. Comprehensive diet management includes the correction of suboptimal dietary factors, patient education in the self-selection of foods that increase compliance with healthy eating and an assessment of how patients relate to food. Patients, especially those with multiple and severe metabolic alterations, as observed in PAD, often have a conflicting relationship with nutrition, which hinders compliance with dietary guidelines and aggravates the severity of individual pathologies and dietary risk factors [684].

Suboptimal eating habits often result from poorly understood eating disorders, which drastically limit the achievement of recommended therapeutic goals and worsen patient outcomes. To the best of our knowledge, there is a gap in the literature on nutrition personalization and the management of eating behaviors which are prevalent in people with PAD. Further studies are needed to evaluate the beneficial effects of specific dietary interventions on patient outcomes, investigating the advantages of each diet on different categories of patients based on individual characteristics, comorbidities and personal needs.

Herein, we reviewed the current knowledge regarding common eating habits and diets, focusing on their impact on the risk factors of PAD. We analyzed the role of nutritional status, the available general recommendations on nutrient supplementation and the complex relationship between PAD and the most frequent prevalent dietary patterns and eating disorders. We studied four diets typical of Western nutritional patterns and composition. Finally, we reviewed and selected the eating behaviors commonly seen in this culture. Our purpose was to provide a comprehensive update on the effect of nutrition on PAD.

Although it was possible to include a wide variety of diets and habits, our goal was to gather the most relevant evidence on nutritional factors with an observed impact on PAD of the lower limbs and associated risk factors. Unfortunately, this fundamental relationship remains undervalued and poorly understood. Therefore, this article provides a unified view of the current knowledge on the topic with an original analysis of an even more unexplored but fundamental aspect of nutrition, namely eating behaviors and related disorders.

We evaluated the impact of each diet and eating disorder on comorbidities and risk factors that frequently affect patients with PAD. Consistent with current evidence, we have confirmed the fundamental role nutrition plays in the primary and secondary prevention of PAD via improving metabolic function, ameliorating the pathologies linked to atherosclerosis and exerting adjuvant properties that increase the success rate of traditional therapies [685].

Particular interest should be devoted to CKD patients with PAD, as they represent a fragile population with limited therapeutic options and have the poorest outcomes compared to groups without CKD [484]. Nutrition in this population also remains a topic to be explored to understand how to prevent the progression of renal disease, with improvement in the quality of life and patient outcomes [488].

The Mediterranean diet is the most studied and suitable dietary model for patients with PAD among the four diets selected in this study. Furthermore, the guidelines recommend adherence to the Mediterranean diet as an additional non-pharmacological strategy to reduce the incidence of atherosclerosis and progression of the most common vascular complications. However, compliance with this nutritional model is hindered by its high cost and difficulty in providing its typical components [672]. Hopefully, globalization and the progressive equalization of socioeconomic discrepancies will facilitate adherence to such a healthy diet [674].

Smoking is one of the most important risk factors for PAD and there exists a strong prevalence of smoking among patients with various eating disorders or non-optimal diets [619]. Unfortunately, there is little published evidence as to the explanation for the co-existence of these harmful conditions.

In the third section of the review, we emphasized the possible relationship between nutrition and progression to more advanced stages of PAD, such as CLTI, amputation and the failure of revascularization. The nutritional status of these unfortunate patients deserves more attention to correct any deficiencies (especially micronutrients) or excessive weight gain. In particular, sarcopenia is a disabling condition that increases the rate of MALE and should be considered a negative prognostic factor for post-revascularization success. Diet and eating behaviors/disorders should be included in the routine clinical evaluation of patients with PAD, as appropriate management may further reduce residual risk, even after pursuing the best pharmacological treatment. The management of patients with PAD should include a thorough evaluation of nutrition and possible eating behaviors to provide targeted nutritional interventions, support individual needs based on their medical history and also potentially prevent the underlying causes of the incidence and progression of PAD. Unfortunately, the importance of diet as a therapeutic and preventive strategy in PAD is still underestimated, resulting in little guideline support in diet-based treatment. Furthermore, the professional figure who takes care of the nutritional aspects of patients by correcting deleterious eating behaviors and providing appropriate dietary interventions has never been established. Probably this professional figure should be identified in a multidisciplinary group of experts (such as trained doctors and nurses, medical assistants, nutritionists, psychologists and health coaches) who can take full responsibility for the patient’s nutritional status. Additionally, proper assessment of patients’ nutritional status should include regular outpatient counseling supported by validated screening questionnaires that help detect incorrect eating behaviors and monitor patient compliance, and regularly scheduled medical evaluations for targeted nutritional interventions and tailored diets that maximize results and increase compliance with dietary recommendations and prescriptions [686].

In conclusion, by highlighting the limitations of the available scientific evidence that cannot effectively identify a correct approach to the eating disorders prevalent in PAD, we show that current knowledge cannot be used to support a personalization of diet based on the needs and characteristics of the patients. We tried to demonstrate the counterproductive effect of underestimating such a fundamental aspect that, if further studied, could counter the global health burden represented by PAD and its related comorbidities.

## 5. Conclusions

Dietary risk factors are a growing cause of global morbidity and mortality, with a significant impact on the incidence, progression and complications of PAD, particularly MACE and MALE. A comprehensive assessment of the nutritional patterns and eating behaviors adopted by patients with PAD is one of the most underestimated aspects of managing this disease and should always be included to achieve an effective improvement in patient outcomes.

Nutrition and dietary risk factors are key elements that deserve much more attention; in particular, eating disorders should be explored to reduce the residual risk for both the primary and secondary prevention of PAD, slow down its progression and reduce the rate of CV and lower extremity complications. The population affected by PAD of the lower limbs deserves dedicated nutritional assessments and personalized dietary models based on individual needs and correct eating behaviors for effective management and improved results.

## Figures and Tables

**Figure 1 ijms-23-10814-f001:**
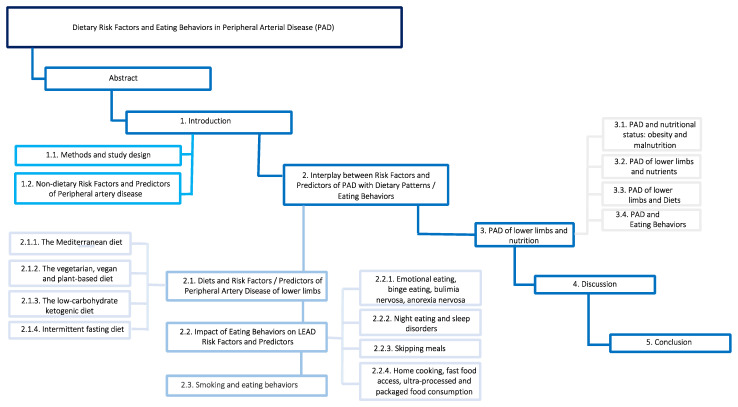
Methods and study design.

**Table 1 ijms-23-10814-t001:** Interplay between non-dietary risk factors and predictors of PAD with diets.

Low Carbohydrate/Ketogenic Diet		
PAD Risk Factors/Predictors	Evidence	Reference
Obesity	Efficient weight loss	[9,10,11,12,13,14,15,16,17,18,19,20]
Diabetes mellitus	Reduction in total insulin requirements due to reduced insulin resistance. Need for further safety studies	[21,22,23,24,25,26,27]
Hypertension	Controversial results with risk of worsening of blood pressure control in particular in CKD	[11,28,29,30,31,32,33,34,35]
Dyslipidemia	Reduction in the size of LDL-c molecules	[11,25,35,36,37,38,39,40,41,42]
Chronic kidney disease	Risk of calcium–phosphate homeostasis disruption, risk of progression of CKD due to acidosis and protein intake. Promotion of cyst regression in polycystic kidney disease	[43,44,45,46]
Inflammation	Modulation of the inflammasome with a net reduction in oxidative stress	[42,47,48,49,50]
**Intermittent fasting diet**		
PAD Risk Factors/Predictors	Evidence	Reference
Obesity	Promotion of weight loss and change in body composition by consuming fat stores. May cause fatigue and dizziness	[51,52,53,54,55,56,57]
Diabetes mellitus	Reduction in serum glucose values regardless of weight loss but increased risk of hypoglycemia	[53,58,59,60]
Hypertension	Promising results by modulating cortisol levels and circadian rhythm. However, further confirmation for safe use in clinical practice is needed	[58,61,62,63,64,65,66,67,68]
Dyslipidemia	Evidence from observational studies on Ramadan. The amelioration of lipid profile results in an overall reduction in cardiovascular risk	[53,69,70,71,72,73,74,75,76,77,78,79]
Chronic kidney disease	Apparently safe in mild/moderate CKD when closely monitored	[80,81,82,83,84,85]
Inflammation	Regression of the systemic inflammatory state by promoting anti-inflammatory molecules	[56,58,77,86,87,88,89,90,91,92,93,94,95,96,97,98]
**Vegetarian and Vegan diet**		
PAD Risk Factors/Predictors	Evidence	Reference
Obesity	Prevention and management of overweight but supplementation of vitamin B12 in the vegan diet is recommended	[99,100,101,102,103,104,105,106,107,108,109,110]
Diabetes mellitus	Improvement of pancreatic beta cell function, and increase in the production of gastrointestinal incretins resulting in a lower total insulin requirement	[99,111,112,113,114,115,116,117,118,119,120,121,122,123]
Hypertension	Additional improvement of the pressure profile if adequate adherence	[124,125,126]
Dyslipidemia	Efficacy comparable to statin therapy	[110,127,128,129,130,131,132]
Chronic kidney disease	Delayed need for dialysis but need to monitor serum potassium and risk of possible malnutrition	[133,134,135,136,137]
Inflammation	Restoration of intestinal microbiota homeostasis (Firmicutes/Bacteroidetes ratio) and reduction in local and systemic inflammation	[138,139,140,141,142,143,144,145,146,147]
**Mediterranean diet**		
PAD Risk Factors/Predictors	Evidence	Reference
Obesity	Multifactorial properties that lead to significant weight loss	[148,149,150,151,152,153,154,155,156]
Diabetes mellitus	Lower intake of high glycemic index foods with a positive impact on glucose management	[148,149,150,151,152,153,154,157,158,159,160,161]
Hypertension	Regression of arterial degenerative processes exerting benefits on endothelial function and stiffness that result in a more effective blood pressure control	[162,163,164,165,166,167,168,169,170,171,172,173]
Dyslipidemia	Reduction in intestinal absorption and endogenous production of cholesterol along with epigenetic control on lipid metabolism	[174,175,176,177,178,179,180,181,182,183,184,185,186,187,188]
Chronic kidney disease	Not effective in preventing the accumulation of GDUTS but promising results in kidney transplantation and in the prevention of kidney stones	[189,190,191,192,193,194,195,196,197,198,199,200,201,202,203,204,205]
Inflammation	“Hormetic therapy” with multilevel regulation of the inflammatory process and homeostasis of the immune system	[158,166,186,197,206,207,208,209,210,211,212,213,214,215,216,217,218,219,220,221,222,223,224,225,226]

**Table 2 ijms-23-10814-t002:** Interplay between non-dietary Risk factors and Predictors of PAD with diets.

Emotional Eating, Binge Eating, Bulimia Nervosa, Anorexia Nervosa		
PAD Risk Factors/Predictors	Evidence	Reference
Obesity	Maladaptive response to distress and discomfort lead to recurrent episodes of overeating resulting in a net weight gain	[227,228,229,230,231,232,233,234,235]
Diabetes mellitus	Harmful compensatory behaviors due to emotional involvement in relation to food that lead to low compliance with therapy, high risk of hypoglycemia and worsening of glucose plasma level control	[236,237,238,239,240,241,242,243,244,245,246]
Hypertension	The emotional substrate directly regulates blood pressure values hindering the achievement of blood pressure targets	[247,248,249,250,251,252]
Dyslipidemia	Stress decompression through overeating increases cholesterol levels	[249,250,252,253]
Chronic kidney disease	-	-
Inflammation	Each eating disorder exhibits a typical inflammatory pattern that requires further evaluation	[248,254,255,256,257,258,259,260,261]
** Night eating and sleep disorders **		
PAD Risk Factors/Predictors	Evidence	Reference
Obesity	Impairment of food regulation patterns leading to hyperphagia due to hedonic eating	[262,263,264,265,266,267,268,269,270,271]
Diabetes mellitus	Hypoxia activates neuro-hormonal triggers resulting in insulin resistance	[272,273,274,275,276,277,278,279,280,281,282,283,284,285,286,287]
Hypertension	Frequent awakenings during sleep alter the night-time blood pressure profile	[288,289]
Dyslipidemia	U-shaped association between sleep habits and serum lipid concentrations. Hypoxic episodes trigger neuro-adrenocortical hyper-activation that increases cortisol plasma levels	[290,291,292,293,294,295,296,297,298,299,300,301,302]
Chronic kidney disease	-	-
Inflammation	-	-
** Skipping meals **		
PAD Risk Factors/Predictors	Evidence	Reference
Obesity	The uneven distribution of calories throughout the day leads to obesity	[303,304,305]
Diabetes mellitus	Skipping meals randomly, alters glucose sensitivity and worsens HbA1c values	[305,306,307,308,309,310,311,312,313,314,315,316]
Hypertension	Progression of the neuro-hormonal mechanisms underlying hypertension	[305]
Dyslipidemia	Unsafe and ineffective long-term strategy for serum lipid control. Skipping meals worsens post-prandial lipid profile	[304,305,308,317]
Chronic kidney disease	-	-
Inflammation	Dysregulation of intestinal microbiota homeostasis with worrying consequences on systemic atherosclerosis	[308]
** Home cooking, fast food access, ultra-processed and packaged food consumption **		
PAD Risk Factors/Predictors	Evidence	Reference
Obesity	Weight control is supported by a wide selection of foods	[318,319,320,321,322,323,324,325,326,327,328,329,330,331,332,333]
Diabetes mellitus	Promotion of a greater perspective on the impact of food on health, especially in low-income countries, that helps prevent chronic diseases such as DM	[334,335,336,337,338,339,340,341,342,343,344,345,346,347]
Hypertension	Patients with hypertension should be trained with educational cooking programs and avoid foods that worsen blood pressure control (e.g., salt consumption)	[348,349,350,351,352,353,354,355,356]
Dyslipidemia	The spread of low-quality foods has anticipated the incidence of atherosclerosis and related complications with a parallel increase in plasma lipid levels	[321,357,358,359,360,361,362]
Chronic kidney disease	Proper cooking methods could reduce the excessive intake of minerals (e.g., potassium, phosphorus) and proteins frequently contained in fast food, ultra-processed and packaged food	[363,364,365,366,367,368,369,370]
Inflammation	The low-quality of ultra-processed and packaged food was observed to increase oxidative stress and decrease in the reducing power of cells	[371,372,373,374,375,376]
** Smoking ** ** and eating behaviors **		
Evidence		Reference
High prevalence of eating disorders, particularly after quitting smoking due to withdrawal symptoms		[268,377,378,379,380,381,382,383]

**Table 3 ijms-23-10814-t003:** PAD, diets and eating disorders.

LEAD and Diets		
Diet	Evidence	Reference
Low carbohydrate/ketogenic diet	Improvement of the metabolic syndrome and beneficial effect on all common comorbidities/risk factors in the PAD population. This dietary model can be adopted for short periods alternating with the MD	[44,384,385]
Intermittent fasting	Correction of dysbiosis with a reduction in inflammation; prevention of atherosclerotic plaque vulnerability by modulating local inflammation and change in the lipid core of the plaque of peripheral arteries; improvement of liver function and glucose/lipid metabolism with consequent secondary prevention of CV risk; concerns about osteoporosis and hypoglycemia in diabetic patients with PAD	[77,386,387,388,389,390]
Vegetarian and Vegan diet	Primary prevention by counteracting the progression of risk factors and comorbidities underlying LEAD development; effect similar to a reversal of the atherosclerotic process underlying the deterioration of the peripheral arteries; antioxidant protection on the inflammatory load of PAD; reduction in the incidence of MACE	[391,392,393,394,395,396,397,398,399,400,401]
Mediterranean diet	Probably the most effective and safe dietary model to adopt in order to prevent the incidence of PAD through the improvement of all coexisting risk factors; additional benefit on secondary prevention along with pharmacological therapies	[398,402,403,404,405]
** PAD and Eating disorders **		
LEAD Risk Factors/Predictors	Evidence	Reference
Emotional eating, binge eating, bulimia nervosa, anorexia nervosa	Patients with PAD and psychiatric disorders involving the emotional sphere have the highest mortality rate due to the progression of the atherosclerotic process underlying PAD; eating disorders including over-eating and malnutrition are prevalent in advanced stages of PAD	[247,406,407,408,409,410,411,412,413,414]
Night eating and sleep disorders	Progression of the clinical stages of PAD; however, eating disorders involving sleep quality are rarely investigated	[415,416,417,418,419,420,421,422,423,424,425]
Skipping meals	Increased morbidity and mortality of patients with PAD by deteriorating the metabolic homeostasis	[308,315,426,427,428,429,430,431,432,433]
Home cooking, fast food access, ultra-processed and packaged food consumption	Dedicated educational cooking programs should be encouraged to prevent and even correct the deleterious dietary patterns often adopted by LEAD patients. The physical and functional impairment experienced by patients in the advanced stages of the disease hinders the proper provision/cooking of healthy foods	[434,435]

**Table 4 ijms-23-10814-t004:** CTLI progression, nutrition, dietary models and eating behaviors/disorders.

CTLI Progression, Nutrition, Dietary Models and Eating Behaviors/Disorders		
Nutritional status and PAD	Evidence	Reference
Obesity	Obese patients with LEAD experience an overall approximately 1.5-fold increase in the development of CTLI regardless of other confounding factors	[436]
Malnutrition and sarcopenia	Malnutrition and sarcopenia have a devastating effect on patient outcomes with a low success rate in foot wound healing, a higher incidence of MACE and major amputations even with appropriate drug treatment. Malnutrition and sarcopenia have a strong prognostic role in limb salvage	[394,437,438,439,440,441,442]
Nutrients and PAD	Evidence	Reference
Vitamins	Vitamin deficiency (especially vitamin B12, C and D) appears to accelerate progression to CTLI, increase the rate of infections (particularly in PAD patients on dialysis), worsen post-amputation outcomes and increase the incidence of MACE and MALE	[443,444,445,446,447,448]
Micronutrients (zinc, magnesium)	Supplements to reduce the risk of progression to CTLI, incidence of revascularization failure or amputation are not supported by evidence	[448]
PUFAs	Eicosapentaenoic acid deficiency might be correlated with a higher incidence of MALE and MACE	[449,450]
Synthetic Fibrate	Fenofibrate in patients with diabetes mellitus further reduces the residual risk of microvascular complications including lower extremity amputation (LEA)	[451,452,453,454,455]
Diet and PAD	Evidence	Reference
Low carbohydrate/ketogenic diet	-	
Intermittent fasting	-	
Vegetarian and Vegan diet	This dietary model is rich in nutrients (especially PUFAs) that reduce the morbidity and mortality of patients with PAD and slow the progression to the advanced stages of the disease	[392,401]
Mediterranean diet	The MD may offer significant benefits in terms of lower mortality and morbidity with protective effects on MACE and MALE. However, the population with PAD often has socioeconomic limitations that reduce adherence to MD resulting in a higher amputation rate in these subcategories of low-income patients.	[398,456,457]
Eating behaviors/disorders and PAD	Evidence	Reference
Emotional eating, binge eating, bulimia nervosa, anorexia nervosa	The emotional burden resulting from progressive physical and functional limitation is often associated with eating disorders (emotional and binge eating), which further worsen survival and limb outcomes. Serum adipokines could interact with this complex interaction between emotional substrate, eating disorders and MALE incidence	[408,409,410,411,412,413,458]
Night eating and sleep disorders	Treatment of sleep and eating disorders is a preventative strategy for MACE and MALE	[424,425,459]
Skipping meals	-	
Home cooking, fast food access, ultra-processed and packaged food consumption	Physical and functional barriers to healthy food supplying and preparation resulting from the disabling complications of PAD further increase the risk of progression into CTLI	[434,435,460]

## Data Availability

All data can be found in the following two digital archives: PubMed and Google Scholar.

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
