# Peer review of "Dietary Risk Factors and Eating Behaviors in Peripheral Arterial Disease (PAD)"

_ijms, 2022, doi:10.3390/ijms231810814_

Round 1
Reviewer 1 Report
Dear author
This review aims to investigate the evidence of dietary risk factors and eating behaviors for lower extremity arterial disease, and is very interesting. I think that a minor thing should be revised to be accepted as follow;
LEAD is used as an abbreviation of Lower extremity arterial disease, but the abbreviated word may not be so much used in PUBMED and any other sites. So, I recommend using the word; PAD(peripheral arterial disease) as a common word in this article.
From reviewer.
Author Response
Dear author
This review aims to investigate the evidence of dietary risk factors and eating behaviors for lower extremity arterial disease, and is very interesting. I think that a minor thing should be revised to be accepted as follow;
LEAD is used as an abbreviation of Lower extremity arterial disease, but the abbreviated word may not be so much used in PUBMED and any other sites. So, I recommend using the word; PAD (peripheral arterial disease) as a common word in this article.
Authors’ response:
We thank Reviewer 1 for raising this shareable concern and we totally agree with this comment. We revised the title and paragraphs of the manuscript and used the acronym PAD (Peripheral Arterial Disease) as a common word in the article. We have left the acronym LEAD (Lower Extremity Arterial Disease) only in isolated parts of the manuscript and only as a synonym for PAD, in order to ensure the readability of the manuscript without compromising the content of the text.
Reviewer 2 Report
Present review highlights the data accumulated over 42 years of randomized clinical trials and experimental studies on the relationship between eating habits (different types of diets) and the development of cardiovascular diseases, including atherosclerosis.
The exact molecular mechanism by which dietary intake of a traditional Mediterranean diet or a vegan diet, for example, has a beneficial effect is not known. If the patient follows these diets, then when analyzing the molecular markers responsible for the development of atherosclerosis, you can see an increase in the expression of vasodilators and a decrease in the expression of vasoconstrictors. In addition, when you eat vegetable ingredients, you do not have animal fats, which increase the concentration of cholesterol in the blood. And in general, the energy consumption of the body for the breakdown of animal and plant macromolecules is not comparable. The positive effect of a «correct» diet is not very noticeable in patients with chronic renal failure, as they have a systemic disturbance of mineral homeostasis, leading to vascular calcification. Therefore, in this case, the diet has a supporting effect rather than a «healing» one.
I do not understand the diagram of the article on page 8 (Figure 1). It becomes clear only after scrolling through the entire manuscript. I recommend the authors to redo it. It will be convenient to read the diagram from left to right. For now, the first arrow leads to «results» and not to «abstract». Let it all start with the «abstract» and the boxes highlighted in blue, then the one highlighted in lilac (chapter 2 of the review), and so on. For an instant understanding that this is a review scheme, you can add the numbering of chapters, since their titles already exist in the scheme.
How does a vegan diet affect on adiponectin and leptin secretion? The answer may complement chapter 2.1.2.
Who should guide the patient on the type of nutrition he needs? Should it be a doctor or nutritionist? How will the doctor monitor the patient's diet? As practice shows, many patients, after receiving the recommendations, do not follow them, or do not follow them completely. Answers to these questions can be added to Chapter 4.
Author Response
Reviewer:
Present review highlights the data accumulated over 42 years of randomized clinical trials and experimental studies on the relationship between eating habits (different types of diets) and the development of cardiovascular diseases, including atherosclerosis.
The exact molecular mechanism by which dietary intake of a traditional Mediterranean diet or a vegan diet, for example, has a beneficial effect is not known. If the patient follows these diets, then when analyzing the molecular markers responsible for the development of atherosclerosis, you can see an increase in the expression of vasodilators and a decrease in the expression of vasoconstrictors.
In addition, when you eat vegetable ingredients, you do not have animal fats, which increase the concentration of cholesterol in the blood. And in general, the energy consumption of the body for the breakdown of animal and plant macromolecules is not comparable. The positive effect of a «correct» diet is not very noticeable in patients with chronic renal failure, as they have a systemic disturbance of mineral homeostasis, leading to vascular calcification. Therefore, in this case, the diet has a supporting effect rather than a «healing» one.
Authors’ response:
We appreciated the comments from Reviewer 2 and agree with his meticulous analysis. We evaluated all the beneficial and deleterious effects that were observed in some of the most studied types of diets. In particular, as Reviewer 2 noted, most of the benefits that nutrition can provide to counteract atherosclerosis and its complications are still unknown and that is why we consider this topic to be crucial for a high-quality level of management of our patients. In particular, Reviewer 2 mentioned the effect of MD and plant-based diet on the expression of vasoconstrictors and vasodilators and, as reported in the manuscript, these two diet models could be directly responsible for increased nitric oxide production, reduced production of endothelin-1, and a modulation of the adrenergic nervous system with even an improvement in the symptoms of claudication in PAD. Additionally, Reviewer 2 highlighted the importance of dedicated nutrition studies in chronic renal failure, as this growing population often deserves specific treatments and also a much more special focus on nutrition. A better understanding of the role and effects of diet on atherosclerosis, its risk factors and its complications, can support the management of PAD, hopefully providing targeted nutritional interventions.
Reviewer:
I do not understand the diagram of the article on page 8 (Figure 1). It becomes clear only after scrolling through the entire manuscript. I recommend the authors to redo it. It will be convenient to read the diagram from left to right. For now, the first arrow leads to «results» and not to «abstract». Let it all start with the «abstract» and the boxes highlighted in blue, then the one highlighted in lilac (chapter 2 of the review), and so on. For an instant understanding that this is a review scheme, you can add the numbering of chapters, since their titles already exist in the scheme.
Authors’ response:
We thank Reviewer 2 for the suggestion and have modified the figure accordingly following his recommendations. The figure now has a horizontal structure. The main chapters are connected in series while the subparagraphs have been depicted with different colors and are linked to the relevant chapters without compromising the linear succession and the development of the main paragraphs of the manuscript.
Reviewer:
How does a vegan diet affect on adiponectin and leptin secretion? The answer may complement chapter 2.1.2.
Authors’ response:
We thank Reviewer 2 for the interesting question that allowed us to investigate this aspect and to further contribute to improving our manuscript. We have added a brief explanation of the effect of a vegan diet on adiponectin and leptin secretion which you can find in chapter 2.1.2. lines 361-372. We revised the manuscript by adding the new citations.
Reviewer:
Who should guide the patient on the type of nutrition he needs? Should it be a doctor or nutritionist? How will the doctor monitor the patient's diet? As practice shows, many patients, after receiving the recommendations, do not follow them, or do not follow them completely. Answers to these questions can be added to Chapter 4.
Authors’ response:
We thank Reviewer 2 for the suggestion. Unfortunately, a professional figure who should take care of the nutritional status of patients has never been established and, as Reviewer 2 noted, this condition results in a noticeable low compliance with dietary recommendations by patients. This is a topic that should deserve much more interest from the scientific community as it could be the first step towards a truly global management of the PAD population. We have added a small comment on this aspect to the manuscript which you can find in chapter 4 lines 1415-1431. Furthermore, we have tried to describe this professional figure and how a nutritional assessment should be structured. We revised the manuscript by adding the new citations.